# VGLL3 is a mechanosensitive protein that promotes cardiac fibrosis through liquid–liquid phase separation

Yuma Horii [1,2,8], Shoichi Matsuda[2,8], Chikashi Toyota[2], Takumi Morinaga[2], Takeo Nakaya[3], Soken Tsuchiya[4], Masaki Ohmuraya[5], Takanori Hironaka[1,2], Ryo Yoshiki[1,2], Kotaro Kasai[1,2], Yuto Yamauchi[1,2], Noburo Takizawa[1,2], Akiomi Nagasaka[2], Akira Tanaka[3], Hidetaka Kosako [6] & Michio Nakaya [1,2,7] ✉

Myofibroblasts cause tissue fibrosis by producing extracellular matrix proteins, such as collagens. Humoral factors like TGF-β, and matrix stiffness are important for collagen production by myofibroblasts. However, the molecular mechanisms regulating their ability to produce collagen remain poorly characterised. Here, we show that vestigial-like family member 3 (VGLL3) is specifically expressed in myofibroblasts from mouse and human fibrotic hearts and promotes collagen production. Further, substrate stiffness triggers VGLL3 translocation into the nucleus through the integrin β1-Rho-actin pathway. In the nucleus, VGLL3 undergoes liquid-liquid phase separation via its low-complexity domain and is incorporated into non-paraspeckle NONO condensates containing EWS RNA-binding protein 1 (EWSR1). VGLL3 binds EWSR1 and suppresses miR-29b, which targets collagen mRNA. Consistently, cardiac fibrosis after myocardial infarction is significantly attenuated in *Vgll3*-deficient mice, with increased miR-29b expression. Overall, our results reveal an unrecognised VGLL3-mediated pathway that controls myofibroblasts' collagen production, representing a novel therapeutic target for tissue fibrosis.

Fibrosis, the pathological deposition of extracellular matrix (ECM) proteins like collagens in the interstitium, can occur in almost all tissues[1–3]. During tissue injury, fibrosis is beneficial to the body as it rapidly compensates for tissue damage. However, excessive fibrosis caused by chronic inflammation or aging results in tissue dysfunction[2]. Fibrosis is involved in approximately 45% of all mortalities in developed countries[3], but there are currently few effective therapies for fibrosis, and new therapeutic agents are needed for its alleviation.

Myofibroblasts are responsible for excessive ECM synthesis and deposition in fibrotic tissues. Myofibroblasts are rare in normal tissues

and are differentiated from multiple cell types such as resident fibroblasts, upon inflammation[4,5]. Their differentiation is triggered by humoral factors secreted by immune cells at the inflammation site[6], like transforming growth factor (TGF)-β which activates Smad signalling[7,8]. Differentiated myofibroblasts produce excessive collagens, which accumulate in tissues and cause hardening. For example, the elastic modulus of a healthy rat heart is about 10 kPa, but in a fibrotic rat heart, it increases to about 50 kPa, resulting in decreased contractility[9]. Myofibroblasts recognise collagen-induced matrix stiffness through receptors like integrins, which activate Rho/Rho-associated protein

[1]Department of Disease Control, Kyushu University, Fukuoka, Japan. [2]Department of Pharmacology and Toxicology, Graduate School of Pharmaceutical Sciences, Kyushu University, Fukuoka, Japan. [3]Department of Pathology, Jichi Medical University, Tochigi, Japan. [4]Department of Pharmaceutical Biochemistry, Graduate School of Pharmaceutical Sciences, Kumamoto University, Kumamoto, Japan. [5]Department of Genetics, Hyogo College of Medicine, Hyogo, Japan. [6]Division of Cell Signaling, Fujii Memorial Institute of Medical Sciences, Tokushima University, Tokushima, Japan. [7]AMED-PRIME, Japan Agency for Medical Research and Development, Tokyo, Japan. [8]These authors contributed equally: Yuma Horii, Shoichi Matsuda. ✉e-mail: nakaya@phar.kyushu-u.ac.jp

kinase (ROCK) signalling and actin polymerisation[10–12]. Actin cytoskeletal formation promotes the expression of ECM proteins[13,14]. Thus, hardening plays an important role in maintaining myofibroblast differentiation, and generates feedback loops that accelerate collagen deposition[15]. However, the molecular mechanisms by which matrix stiffness increases fibrosis-related gene expression in myofibroblasts remain largely unknown.

Liquid–liquid phase separation (LLPS) is a novel phenomenon that explains the dynamic binding of molecules containing RNA-binding proteins to membrane-free organelles and condensed bodies[16]. Condensates containing Non-POU domain containing octamer binding protein (NONO), an RNA-binding protein, are one of the biomolecular condensates formed by LLPS. Among the NONO condensates, the most well-known one is a paraspeckle, which is assembled from various RNA-binding proteins via LLPS, using an architectural lncRNA called nuclear paraspeckle assembly transcript 1 (Neat1) as the backbone[17–20]. Paraspeckles have been reported to have various physiological functions and cause various pathological conditions[17,20–22]. On the other hand, it has been recently shown that non-paraspeckle NONO condensates also have biological functions[18,19]. However, the contribution of phase-separated condensates including NONO condensates to the pathogenesis of fibrosis remains unclear.

Vestigial-like family member 3 (VGLL3) is a transcription cofactor[23–25] that binds TEA Domain (TEAD) transcription factors[24]. VGLL3 is involved in myogenesis[25], cell proliferation[26] and female-biased autoimmune diseases[27]. However, the physiological and pathological roles of VGLL3 remain poorly understood. In this study, we reveal a role for VGLL3, which is specifically induced in cardiac and hepatic myofibroblasts, in collagen production during fibrosis. We show that VGLL3 is translocated to the nucleus by substrate stiffness and undergoes LLPS, which promotes collagen production.

## Results

### Substrate stiffness dictates myofibroblast differentiation in vitro

During our previous study[28], we accidentally found that cardiac myofibroblasts can be de-differentiated simply by culturing in suspension for several days, i.e. by removing physical stimuli. Further, we found that de-differentiated cells can re-differentiate into myofibroblasts only upon physical stimulus and adherent culture for several days. These results were consistent with previous reports demonstrating that myofibroblast differentiation is regulated by the surrounding ECM stiffness[10,15]. We thus hypothesised that comparing the gene expression of these cells, could reveal factors involved in myofibroblast differentiation by substrate stiffness and set up an experiment accordingly. Cardiac myofibroblasts were isolated from fibrotic mouse hearts after myocardial infarction (MI). In brief, we digested the fibrotic mouse hearts using enzymes and removed the erythrocytes. Then, the constituent cells were subjected to overnight culture on plastic plates. After that, the attached cells were collected and subjected to magnetic-activated cell sorting (MACS) separation using anti-CD45 antibody. CD45-negative cells were collected as myofibroblasts. Almost all the collected cells were positive for α-smooth muscle actin (αSMA), a myofibroblast marker protein (Supplementary Fig. 1a, b). The purified cardiac myofibroblasts were cultured on polystyrene plates for 7 days; some of these cells were collected and designated as "adherent" myofibroblasts (Fig. 1a, Supplementary Fig. 1c, d). The remaining myofibroblasts were transferred to ultra-low attachment plates, and cultured in suspension for 7 days, and designated "non-adherent" myofibroblasts (Fig. 1a). These cells almost entirely lacked myofibroblast marker proteins such as αSMA encoded by Acta2 gene and periostin (Fig. 1b). In addition, they lost the expression of fibroblast marker genes (Thy1, Tcf21)[29] (Supplementary Fig. 1e). Instead, they exhibited increased expression of the stem cell marker Oct4 and mesenchymal stem cell marker genes (Islr, Nt5e)[30,31]

(Supplementary Fig. 1e), suggesting their de-differentiation. When non-adherent myofibroblasts were re-plated in a polystyrene plate and cultured for 7 days, myofibroblast marker protein expression in the re-plated cells, designated "re-adherent" myofibroblasts, was restored (Fig. 1a, b). Consistently, the mRNA levels of multiple genes involved in fibrosis were downregulated in "non-adherent" myofibroblasts compared with "adherent" myofibroblasts but were greatly upregulated in "re-adherent" myofibroblasts (Fig. 1c, Supplementary Fig. 1e).

### Vgll3 expression is positively correlated with substrate stiffness-dependent expression of collagen by myofibroblasts

The genes expressed in adherent, non-adherent and re-adherent myofibroblasts were compared using DNA microarrays. We identified 107 genes that were greatly downregulated (over 16-fold, compared with "adherent" myofibroblasts) in de-differentiated myofibroblasts and highly upregulated (over 16-fold, compared with "non-adherent" myofibroblasts) upon re-differentiation (Fig. 1d). In this set, 8 genes showed remarkably increased expression (greater than 2.639-fold) in the mouse heart, 3 days after inducing MI (Fig. 1d). Of these, we focused on the transcription cofactor VGLL3[23–25], because its physiological function is largely unknown.

Vgll3 mRNA levels in myofibroblasts were dramatically decreased upon de-differentiation by suspension culture and were greatly increased upon re-differentiation by re-adherence (Fig. 1e). Among other VGLL family genes[24] (Vgll1, Vgll2 and Vgll4), Vgll4 was expressed in cardiac myofibroblasts (Supplementary Fig. 1f) but the expression of this gene was much lower compared to Vgll3 (Supplementary Fig. 1g).

We then examined whether Vgll3 expression was affected by substrate stiffness in myofibroblasts from other organs and species. Culture of the human interstitial myofibroblast cell line CCD-18Co[32] in suspension for 7 days led to decreased expression of VGLL3, as well as fibrotic genes, whereas substrate stiffness restored the expressions of these genes (Supplementary Fig. 1h, i), as in mouse cardiac myofibroblasts. Expression of Vgll3 mRNA in myofibroblasts or VGLL3 mRNA in CCD-18Co is positively correlated with mRNA expression of collagen regulated by substrate stiffness (Supplementary Fig. 1j).

### VGLL3 cellular localisation is regulated by substrate stiffness

As VGLL3 is reported to be a transcriptional coactivator that binds to TEAD proteins[24–26], we examined its cellular localisation in mechanically stimulated and unstimulated myofibroblasts. Anti-VGLL3 immunostaining revealed that VGLL3 was primarily located in the nucleus in adherent myofibroblasts (Fig. 1f, g). Conversely, in 2 h suspension cultures, VGLL3 expression was predominantly cytoplasmic (Fig. 1f, g). In these experiments, YAP1, a transcriptional coactivator translocated to the nucleus by substrate stiffness[33], was used as a positive control (Supplementary Fig. 2a, b). Similar results were obtained with cardiac myofibroblasts expressing exogenous FLAG-tagged VGLL3 in the same experiment (Supplementary Fig. 2c, d). In cardiac myofibroblasts cultured on hydrogel substrates of varying stiffness (1, 8, 25 and 50 kPa), VGLL3 translocated to the nucleus depending on the intensity of mechanical stimulus (Fig. 1h, i) as did YAP1 (Supplementary Fig. 2e, f). A time-course analysis of VGLL3 nuclear translocation in cardiac myofibroblasts after attachment to glass-bottom plates revealed that VGLL3 was primarily cytoplasmic after 2 h culture in suspension but was rapidly recruited to the nucleus after attachment, and at 4 h after attachment, VGLL3 was predominantly nuclear (Fig. 1j, Supplementary Fig. 2g). Consistent with these data, EGFP-VGLL3, retrovirally expressed in myofibroblasts (Fig. 1k) in fibrotic regions where the tissue stiffness is increased by the accumulation of collagen (Supplementary Fig. 2h), was localised to the nucleus. In contrast, EGFP control was uniformly present in the nucleus and cytoplasm of the myofibroblasts at the fibrotic regions (Supplementary Fig. 2i). These observations indicate that VGLL3 nuclear translocation depends on the intensity of matrix stiffness to myofibroblasts.

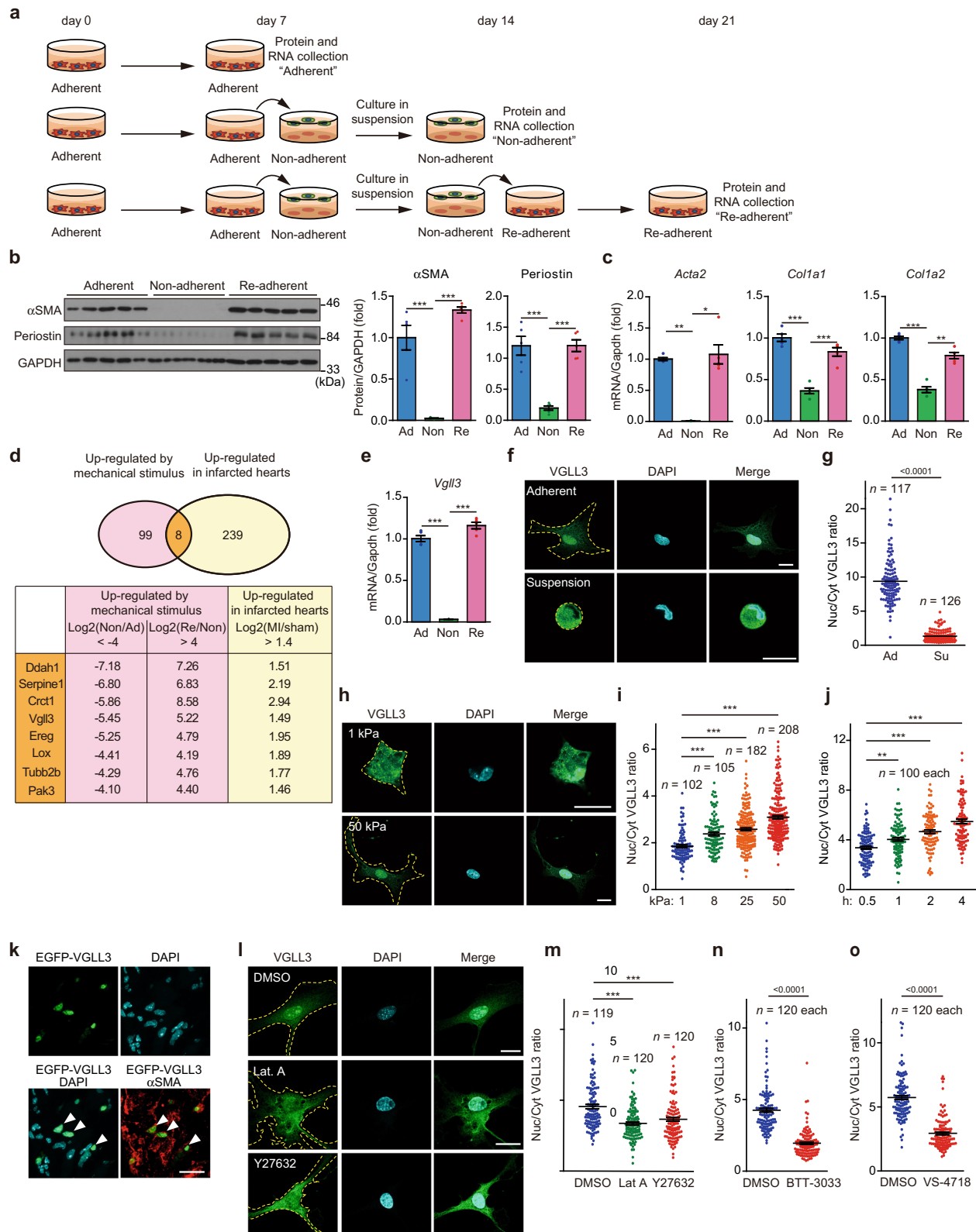

## VGLL3 nuclear translocation by substrate stiffness is regulated by actin polymerisation, downstream of Rho/ROCK and integrin β1 signalling

Actin cytoskeletal tension is the major intracellular mediator of mechanical stimuli[34]. Therefore, we examined whether VGLL3 nuclear translocation upon substrate stiffness was dependent on filamentous actin (F-actin). Adherent myofibroblasts were thus treated with the actin polymerisation inhibitor latrunculin A and analysed for VGLL3 localisation. Latrunculin A increased the cytoplasmic VGLL3 localisation and decreased nuclear VGLL3 localisation (Fig. 1l, m). Blebbistatin, a myosin inhibitor, also reduced VGLL3 nuclear localisation in adherent myofibroblasts (Supplementary Fig. 2j, k). Actomyosin function thus appears necessary for VGLL3 nuclear localisation induced by substrate stiffness.

**Fig. 1 | Substrate stiffness induces *Vgll3* expression and its nuclear translocation in myofibroblasts. a** Schematic representation of the protocol for regulating myofibroblast differentiation using substrate stiffness. **b** Western blot detection of αSMA, periostin and GAPDH in Adherent (Ad), Non-adherent (Non), and Re-adherent (Re) cardiac myofibroblasts ($n = 5$ each). **c** mRNA levels of fibrosis-related genes in Ad, Non and Re cardiac myofibroblasts ($n = 5$ each). **d** Venn diagram of genes upregulated by substrate stiffness and in infarcted hearts. **e** mRNA levels of *Vgll3* in Ad, Non, and Re cardiac myofibroblasts ($n = 5$ each). **f–i** Confocal immunofluorescence images of endogenous VGLL3 in myofibroblasts cultured in suspension or adherent culture (**f**), or plated on soft (1 kPa) and stiff (50 kPa) hydrogels (**h**). Graphs indicate the ratios of nuclear to cytoplasmic VGLL3 intensities in myofibroblasts cultured in adherent (Ad) or suspension (Su) culture (**g**), or plated on hydrogels with different elastic modulus (**i**). **j** Ratios of nuclear to cytoplasmic VGLL3 intensities in myofibroblasts attached for different durations on plastic plate. **k** Nuclear localisation of EGFP-VGLL3 in myofibroblasts (αSMA) in the left ventricle of MI murine hearts on day 3. White arrowheads in merged images indicated representative signals for VGLL3 in myofibroblasts. **l** Confocal immunofluorescence images of myofibroblasts treated with DMSO (0.5%), latrunculin A (Lat A, 2 μM), and Y27632 (80 μM). **m–o** Ratios of nuclear to cytoplasmic VGLL3 intensities in myofibroblasts treated with the cytoskeletal inhibitors (**m**), the integrin β1 inhibitor BTT-3033 (30 μM) (**n**), or FAK inhibitor VS-4718 (50 μM) (**o**). All experiments were performed at least three times. Data in (**b**, **c**, **e**, **g**, **i**, **j**), and (**m–o**) are shown as the mean ± SEM. *P*-values were determined using one-way ANOVA followed by Tukey's range test in (**b**, **c**) (*Col1a1* and *Col1a2*) and (**e**), Kruskal–Wallis followed by Dunn's test in **c** (*Acta2*), (**i**, **j**, **m**), and two-sided Mann–Whitney's *U* test in **g**, **n**, and **o**, *$P < 0.05$, **$P < 0.01$, ***$P < 0.001$. Scale bars in (**f**, **h**, **k**) and **l** = 20 μm. Source data are provided as a Source Data file.

Rho/ROCK signalling, a major pathway that induces actin polymerisation during mechanotransduction[11], was then examined in the VGLL3 nuclear translocation upon substrate stiffness. Treatment of cardiac myofibroblasts with the ROCK inhibitor Y27632 (Fig. 1l, m) or the Rho inhibitor cell-permeable C3 transferase (Supplementary Fig. 2j, k) significantly reduced VGLL3 nuclear localisation, similar to latrunculin A or blebbistatin, demonstrating Rho/ROCK pathway involvement in mechanically induced VGLL3 nuclear translocation.

Integrin β1 senses ECM stiffness by recognising collagen accumulation in fibrotic tissue and triggers intracellular signalling that promotes fibrosis[10,11,35,36]. Treatment with the integrin β1 inhibitor BTT-3033[37] significantly reduced VGLL3 nuclear translocation, indicating the involvement of integrin β1 in this process (Fig. 1n, Supplementary Fig. 2l). Further, treatment with VS-4718[38,39], an inhibitor of FAK, which is a focal adhesion component that links integrins and the F-actin cytoskeleton[40], significantly decreased VGLL3 nuclear localisation (Fig. 1o, Supplementary Fig. 2m). These results indicate that the integrin β1/FAK pathway is involved in VGLL3 nuclear translocation by substrate stiffness.

## *Vgll3* expression is induced specifically in myofibroblasts in mouse and human fibrotic heart after myocardial infarction

A time-course analysis was performed to determine the correlation of *Vgll3* mRNA levels with fibrosis progression in mouse hearts after inducing MI (by permanent occlusion of the left anterior descending artery). The results revealed that the mRNA levels of fibrotic genes, such as collagens and *Periostin* (*Postn*) were greatly increased in the infarct area of the mouse heart after MI (Fig. 2a, Supplementary Fig. 3a), peaked at 7 days after MI and then gradually decreased; however, the levels in the infarcted area remained high even on day 28 after MI. Similar time-dependent changes after MI were found for the mRNA expression for *Vgll3* (Fig. 2a), suggesting that *Vgll3* mRNA expression is associated with fibrosis. We next examined the *Vgll3* expression in cells other than myofibroblasts in infarcted mouse hearts by in situ hybridisation, because there are no anti-VGLL3 antibodies including our custom VGLL3 one applicable for immunohistochemistry. In situ hybridisation analysis of the mouse heart, 7 days after MI revealed that *Vgll3* mRNA was primarily expressed in the infarcted area. Furthermore, *Vgll3* mRNA signals were detected in myofibroblasts that expressed αSMA (Fig. 2b) or *Postn* mRNA (Supplementary Fig. 3b), but not in cardiomyocytes (α-actinin–positive cells) (Fig. 2c) and leucocytes (CD45-positive cells) (Fig. 2d). Myofibroblast-specific *Vgll3* expression was also confirmed in mouse hearts at 28 days after MI (Supplementary Fig. 3c, d). In contrast, *Vgll3* mRNA signals were absent in sham-operated control hearts (Supplementary Fig. 3e) with few myofibroblasts. Consistently, analysis of the GEO dataset (GSE116250) including patients with heart failure (ischaemic cardiomyopathy and dilated cardiomyopathy)[41] manifesting cardiac fibrosis and control individuals demonstrated that *VGLL3* mRNA expression was significantly upregulated in the fibrotic heart (Fig. 2e). Further, in situ hybridisation analysis of human heart specimens after autopsy revealed that *VGLL3* mRNA signals were found in the fibrotic area of the hearts of patients with MI and *VGLL3* expression was specifically found in cardiac myofibroblasts (Fig. 2f). Conversely, the hearts of non-MI patients did not show *VGLL3* expression (Supplementary Fig. 3f). Thus, *Vgll3* is selectively expressed in the myofibroblasts of fibrotic mouse and human hearts after MI, suggesting *VGLL3* involvement in the pathogenesis of MI.

## *Vgll3* expression is induced specifically in myofibroblasts in mouse fibrotic liver

To investigate whether *Vgll3* mRNA expression was also induced in the mouse liver upon fibrosis, mice were treated with carbon tetrachloride ($CCl_4$) for 4 weeks to induce hepatic injury, inflammation and subsequent fibrosis (Supplementary Fig. 4a). As the liver is a highly regenerative organ, the damaged mouse liver was almost entirely regenerated 4 weeks after $CCl_4$ discontinuation, with disappearance of inflammation and fibrosis (Supplementary Fig. 4a). *Vgll3* expression was correlated with the *Col1a1* and *Col1a2* expression (Supplementary Fig. 4b). Similar to the findings in fibrotic infarcted hearts (Fig. 2b), *Vgll3* was expressed in activated hepatic stellate cells corresponding to liver myofibroblasts, but not in hepatocytes or Kupffer cells in fibrotic livers (Supplementary Fig. 4c). *Vgll3* expression was also increased in the fibrotic liver of non-alcoholic steatohepatitis (NASH) model mice (Supplementary Fig. 4d), specifically in hepatic stellate cells expressing Desmin (Supplementary Fig. 4e). Analysis of a GEO dataset (GSE126848) including 16 patients with NASH and 14 control patients[42] showed that *VGLL3* expression was significantly upregulated in the fibrotic livers of patients with NASH (Supplementary Fig. 4f). These observations suggest that *Vgll3* expression is induced in tissues other than the heart upon fibrosis progression, specifically in myofibroblasts.

## *Vgll3* expression is induced both by substrate stiffness depending on actin polymerisation/integrin β1 and by TGF-β stimulation

We next examined the mechanism regulating *Vgll3* mRNA expression in myofibroblasts by substrate stiffness (Fig. 1). As actin cytoskeletal formation is important for maintaining myofibroblast differentiation induced by substrate stiffness[43], we treated cardiac myofibroblasts with latrunculin A, BTT-3033, and CCG-1423, a Rho/MRTF/SRF pathway inhibitor. We found that treatment with these inhibitors significantly decreased the *Vgll3* mRNA levels in myofibroblasts, similar to *Acta2*, indicating that *Vgll3* expression induced by substrate stiffness in myofibroblasts was regulated by formation of the actin cytoskeleton, downstream of integrin β1 (Fig. 2g–i).

Although we focused on substrate stiffness-induced *Vgll3* expression, *Vgll3* was rapidly induced even in the mouse heart, where there are few ECM proteins that generate matrix stiffness, upon myocardial infarction (Figs. 2a, 3 days after MI). Upon tissue injury, myofibroblasts are initially differentiated from resident

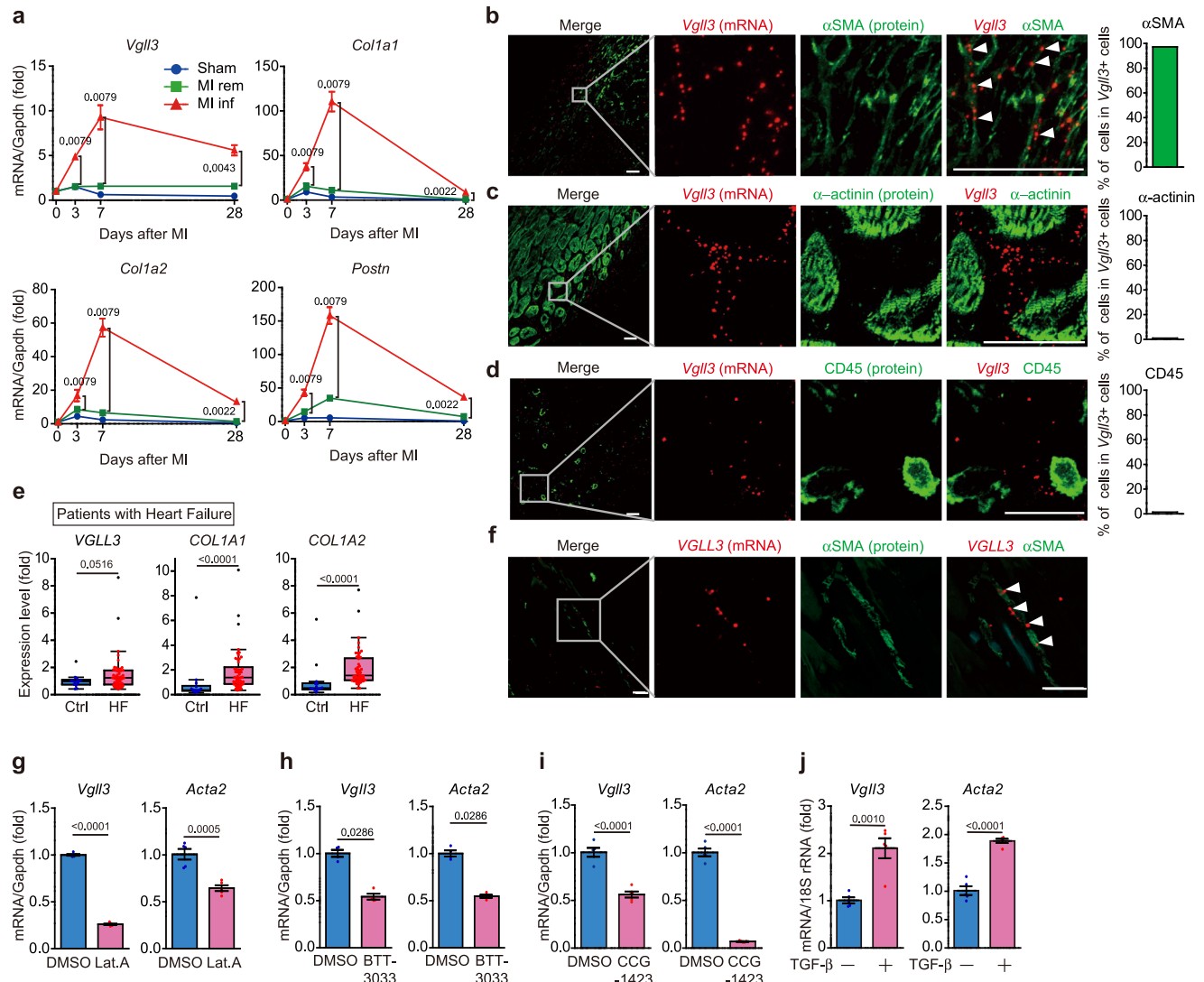

**Fig. 2 | *Vgll3* is induced in fibrotic heart and is specifically expressed in myo-fibroblasts. a** mRNA levels of *Vgll3* and fibrosis-related genes in sham-operated mouse hearts (sham), and in the remote (rem) and infarcted (inf) areas of mouse hearts after myocardial infarction (MI) ($n = 3$–6 hearts/group). **b**–**d** Co-detection of *Vgll3* mRNA and αSMA (**b**), α-actinin (**c**) and CD45 (**d**) in the left ventricle of MI murine hearts on day 7. Percentage of cells positive for each marker protein in *Vgll3* + cells is shown in each graph (n > 100 cells). **e** mRNA levels of *VGLL3* and fibrosis-related genes in the hearts of control individuals (Ctrl) ($n = 14$) and patients with heart failure (HF) ($n = 50$) based on data from GSE116250. **f** Co-detection of *VGLL3* mRNA and αSMA in the left ventricle of patients with MI. **g**–**i** *Vgll3* and *Acta2* mRNA levels in cardiac myofibroblasts treated with Lat.A (2 μM) for 4 h ($n = 5$ each) (**g**), BTT-3033 (15 μM) for 12 h ($n = 4$ each) (**h**) and CCG-1423 (10 μM) for 24 h ($n = 5$ each) (**i**). **j** *Vgll3* and *Acta2* mRNA levels in neonatal rat cardiac fibroblasts treated with TGF-β (10 ng/mL) for 24 h ($n = 4$ each). Data in (**e**) are presented as box and whisker plots (Tukey style, outliers in black dots). The box shows the 25th to 75th percentile range with the median value represented by a horizontal line. The whiskers stretch to the minimum and maximum values within 1.5 times the inter-quartile range from the 25th–75th percentiles. Data in (**a**), and (**g**–**j**) are presented as the mean ± SEM. *P*-values were determined using the two-sided Mann–Whitney's *U* test in (**a**, **e**, **h**), two-sided Student's *t* test in (**g**, **i**, **j**). All in situ hybridisation data are representative of at least three independent experiments. White arrowheads in (**b**, **f**) indicate the representative signals for *Vgll3* or *VGLL3* mRNA. Scale bars in **b**–**d**, **f** = 30 μm. Source data are provided as a Source Data file.

fibroblasts[44] by humoral factors, mainly TGF-β, secreted by macro-phages. We thus examined whether *Vgll3* is also induced during trans-differentiation from resident fibroblasts into myofibroblasts by TGF-β stimulation. Consistent with previous results[45], neonatal rat cardiac fibroblasts treated with TGF-β showed increased *Acta2* mRNA levels, indicating their differentiation into myofibroblasts (Fig. 2j). This treatment significantly increased *Vgll3* expression (Fig. 2j), indicating that *Vgll3* expression is also induced by TGF-β-mediated differentiation of myofibroblasts from resident fibroblasts.

**VGLL3 enhances collagen expression in myofibroblasts**

To determine the role of *Vgll3* in cardiac myofibroblasts, we examined the gene expression changes induced by *Vgll3* knockdown in cells. MA plot analysis of the RNA sequencing data revealed that 1314 genes were

decreased (M < −0.5, A > 0) in cardiac myofibroblasts treated with siRNA against *Vgll3* (Fig. 3a). Comprehensive Gene Ontology (GO) analysis using DAVID on these 1314 genes suggested that extracellular matrix genes in the myofibroblasts were downregulated by *Vgll3* knockdown (Fig. 3b). Further, in cardiac myofibroblasts treated with si*Vgll3*, qRT-PCR demonstrated significantly reduced mRNA levels of extracellular matrix genes, *Col1a1, Col1a2, Col6a1, Col14a1* and *Postn* (Fig. 3c, Supplementary Fig. 5a), indicating that VGLL3 increases the mRNA expression of these genes. Consistently, Collagen I protein levels were significantly decreased in cardiac myofibroblasts treated with siRNA against *Vgll3* (Fig. 3d). Conversely, in VGLL3-overexpressing NIH3T3 cells, the mRNA levels of *Col1a1* and *Col1a2* were significantly increased (Fig. 3e). Interestingly, VGLL3 over-expression also increased the amount of endogenous *Vgll3* mRNA

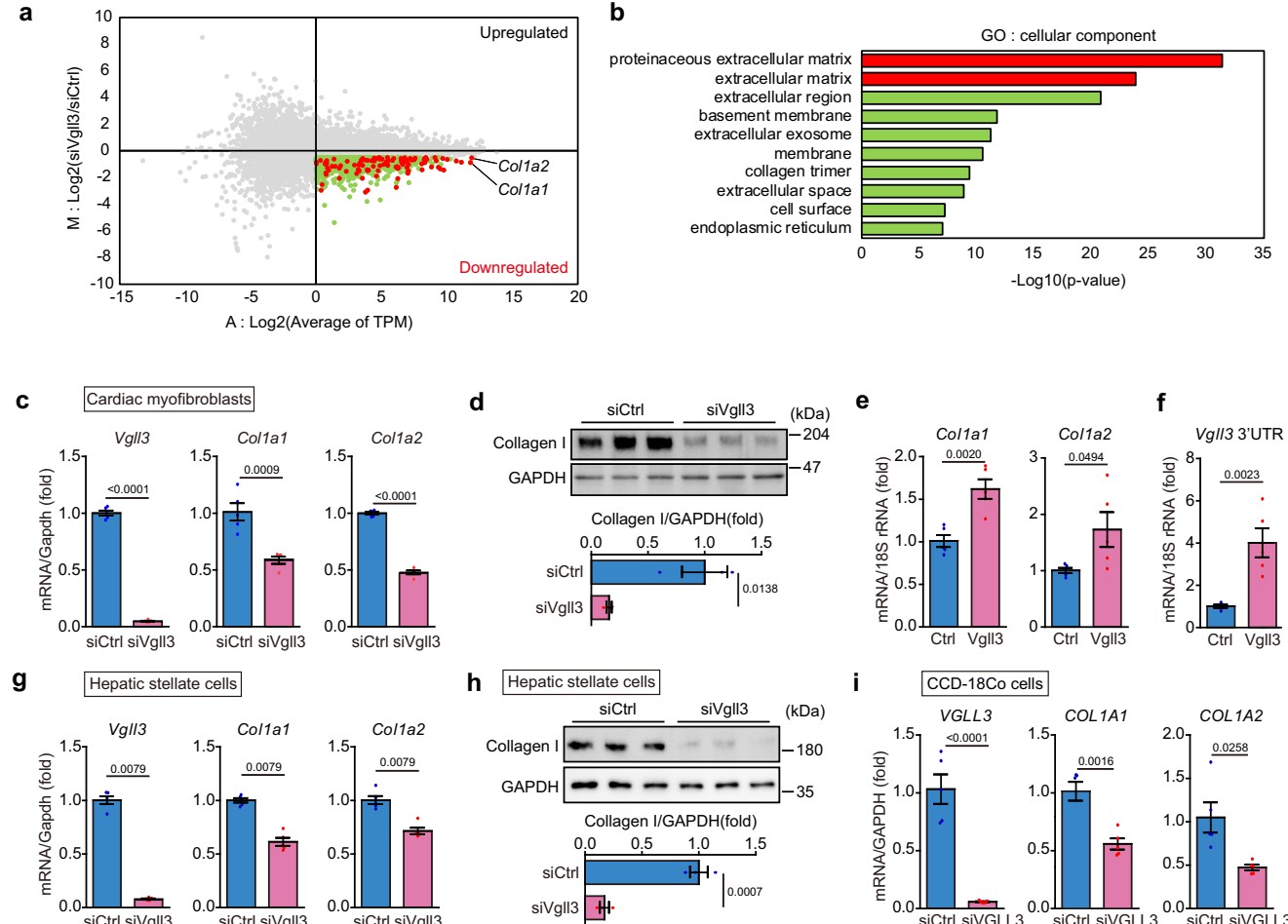

**Fig. 3 | *Vgll3* promotes fibrosis-related gene expression in myofibroblasts.**
**a** Representative MA plot of RNA-seq data demonstrating upregulated or down-regulated genes by si*Vgll3* treatment in cardiac myofibroblasts. Red and green dots represent genes downregulated by si*Vgll3* treatment ($M < -0.5$, $A > 0$). Red and green dots represent proteinaceous extracellular matrix genes/extracellular matrix genes and the other genes shown in (**b**), respectively. **b** Enrichment analysis of ontology for the genes downregulated by si*Vgll3* treatment. **c** mRNA levels of *Vgll3* and fibrosis-related genes in cardiac myofibroblasts transfected with siRNA (#1) targeting *Vgll3* ($n = 5$ each). **d** Protein levels of Collagen I and GAPDH in cardiac myofibroblasts transfected with siRNA (#1) targeting *Vgll3* ($n = 3$ each). **e** *Col1a1* and *Col1a2* mRNA levels in NIH3T3 cells overexpressing FLAG-VGLL3 ($n = 5$ each).

**f** mRNA levels of *Vgll3* 3′UTR in NIH3T3 cells overexpressing FLAG-VGLL3 ($n = 5$ each). **g** mRNA levels of *Vgll3* and fibrosis-related genes in hepatic stellate cells isolated from the livers of NASH model mice, and transfected with siRNA (#1) targeting *Vgll3* ($n = 5$ each). **h** Protein levels of Collagen I and GAPDH in hepatic stellate cells isolated from the livers of NASH model mice, and transfected with siRNA (#1) targeting *Vgll3* ($n = 3$ each). **i** mRNA levels of *VGLL3* and fibrosis-related genes in CCD-18Co transfected with siRNA targeting *VGLL3* ($n = 5$ each). All experiments were conducted at least three times. Data in (**c**–**i**) are presented as the mean ± SEM. *P*-values were determined using the one-sided Fisher's exact test in (**b**), the two-sided Student's *t* test in (**c**–**f**, **h**, **i**), and the two-sided Mann–Whitney's *U* test in (**g**), Source data are provided as a Source Data file.

(3′UTR of *Vgll3* mRNA) in NIH3T3 cells, indicating that VGLL3 promotes the transcription of its own gene (Fig. 3f). Significant reduction of collagen mRNA and Collagen I protein by *Vgll3* knockdown was observed in cardiac myofibroblasts, as well as in liver myofibroblasts isolated from NASH model mice (Fig. 3g, h), and from mice with $CCl_4$-induced liver fibrosis (Supplementary Fig. 5b), or CCD-18Co cells (Fig. 3i, Supplementary Fig. 5c). These results suggest that VGLL3 increases collagen expression in myofibroblasts from various tissues. We further found that knockdown of *Vgll4*, another VGLL family protein that was slightly expressed in cardiac myofibroblasts (Supplementary Fig. 1e), did not significantly influence the *Col1a1* and *Col1a2* levels in myofibroblasts (Supplementary Fig. 5d), suggesting that VGLL4 is not involved in regulating their expressions.

### Identification of VGLL3-interacting proteins by immunoprecipitation-mass spectrometry
To explore the mechanism underlying increased collagen expression by VGLL3, proteins interacting with VGLL3 were identified using immunoprecipitation-mass spectrometry. Isolated cardiac

myofibroblasts retrovirally transduced with constructs encoding FLAG-tagged VGLL3 or a control vector, cultured on polystyrene plates, and processed for immunoprecipitation with anti-FLAG antibodies. Liquid chromatography coupled with tandem mass spectrometry (LC-MS/MS) and label-free quantification revealed multiple VGLL3-binding candidate proteins (Fig. 4a, Supplementary data 1). Among them, we identified the TEAD family member TEAD1, which reportedly interacts with VGLL family proteins[24,25]. The effect of TEAD1 on fibrotic gene expression in cardiac myofibroblasts was thus examined. However, siRNA targeting *Tead1* did not affect fibrotic gene expression in cardiac myofibroblasts (Supplementary Fig. 6), suggesting that TEAD1 is not involved in the VGLL3-mediated fibrotic pathway.

### VGLL3 can undergo phase separation
Unexpectedly, mass spectrometry analysis identified many RNA-binding proteins as candidate VGLL3 interactors (Fig. 4a, b). Among them, multiple RNA-binding proteins harbour a Prion-like domain and are prone to undergo LLPS, including RBM14, TAR DNA-Binding

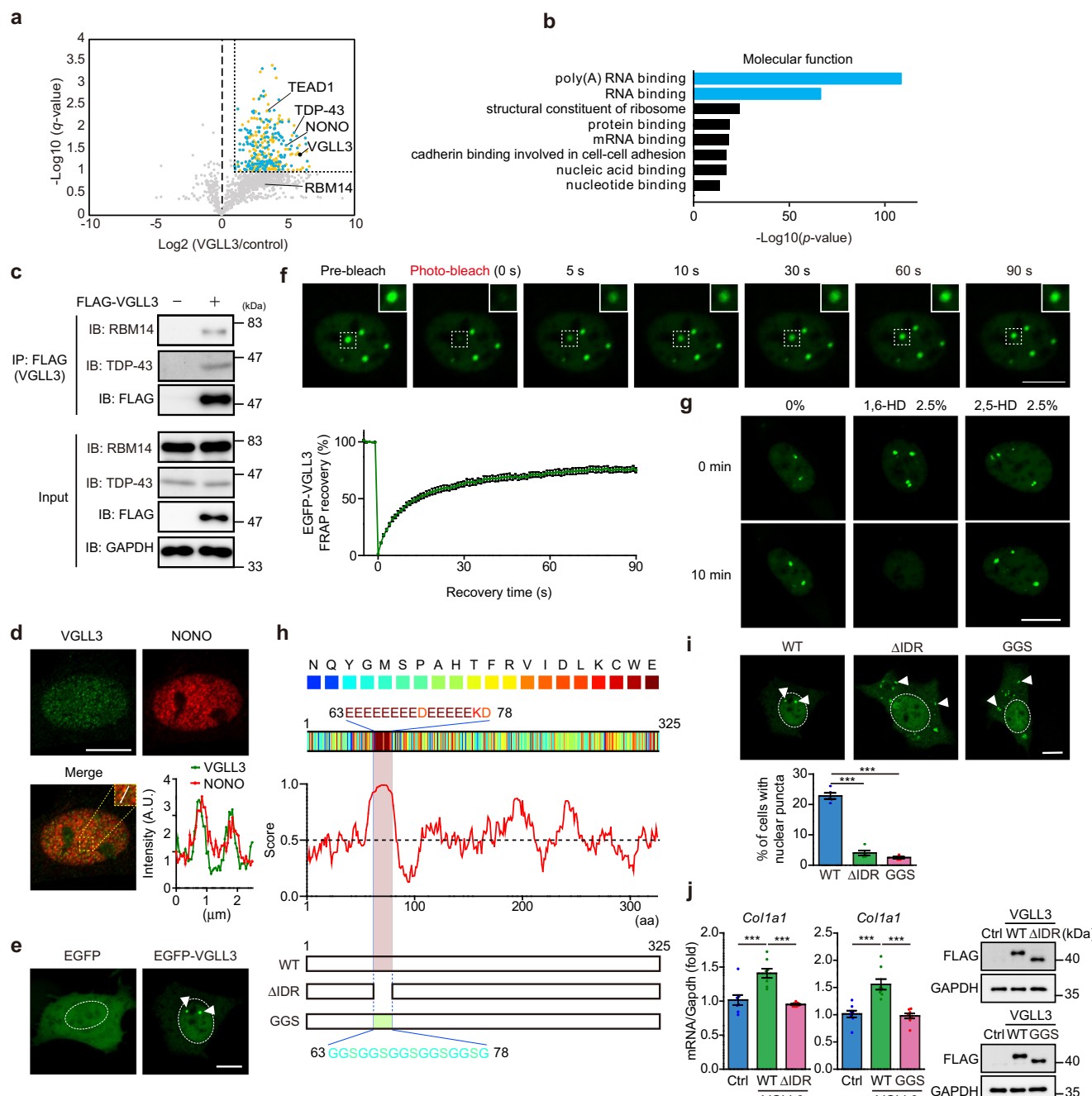

**Fig. 4 | VGLL3 undergoes liquid–liquid phase separation through its glutamic acid-rich low-complexity domain. a** Volcano plot of differential protein profiles in anti-FLAG immunoprecipitates from control and FLAG-VGLL3–overexpressing myofibroblasts. Orange and blue dots represent the proteins ($q < 0.1$, fold increase $>2$). Blue dots represent poly-(A) RNA binding and RNA binding proteins. **b** Enrichment analysis of the ontology for proteins that significantly interact with FLAG-VGLL3. **c** Interaction between FLAG-VGLL3 and endogenous RBM14 and TDP-43 in cardiac myofibroblasts. **d** Immunostaining of endogenous VGLL3 and NONO in myofibroblasts. The graph represents line scans along the white line in the yellow dashed square in the inset. **e** Images of live NIH3T3 cells overexpressing EGFP or EGFP-VGLL3. **f** Fluorescence recovery after photobleaching (FRAP) analysis of EGFP-VGLL3 puncta in live NIH3T3 cells. The mean intensity of normalised fluorescence was shown in the graph ($n = 15$). **g** Effects of 1,6-HD and 2,5-HD treatments on EGFP-VGLL3 puncta in NIH3T3 cells. **h** Scheme of the primary sequence of mouse VGLL3, with individual amino acids colour-coded by the PLAAC algorithm. Schematic prediction of intrinsic disorder tendency in mouse VGLL3 by IUPred2A. The intrinsically disordered region (IDR; aa 63–78) in mouse VGLL3 is highlighted. All amino acid residues in the IDR region were deleted (ΔIDR) or replaced with glycine (G) and serine (S) residues (GGS) in VGLL3 ΔIDR mutant or GGS mutant, respectively. **i** Images of live NIH3T3 cells overexpressing EGFP-VGLL3 (WT) and EGFP-VGLL3 mutants (ΔIDR and GGS). The graph represents the percentage of cells harbouring nuclear puncta in the cells ($n = 5$). **j** *Col1a1* mRNA levels in the cells ($n = 8$). The protein levels of overexpressed FLAG-VGLL3s in the cells were evaluated by western blotting. All experiments were performed at least three times. Data in (**i**, **j**) are shown as the mean ± SEM. *P*-values were determined using one-way ANOVA followed by Tukey's range test in (**i**, **j**), ***$P < 0.001$. White dashed circles or white arrowheads in (**e**, **i**) mark the nucleus or the representative puncta, respectively. Scale bars = 10 μm. Source data are provided as a Source Data file.

Protein 43 (TDP-43) and NONO[17]. To confirm the interaction between these RNA-binding proteins and VGLL3, lysates from cardiac myofibroblasts expressing FLAG-tagged VGLL3 were immunoprecipitated with anti-FLAG antibodies, followed by western blotting with antibodies specific for RBM14 and TDP-43 (Fig. 4c). These experiments revealed that endogenous RBM14 and TDP-43 interact with FLAG-tagged VGLL3. Immunocytochemical analysis for the co-localisation of VGLL3 and NONO in cardiac myofibroblasts revealed that endogenous VGLL3 formed puncta in the nucleus, and these were co-localised with endogenous NONO (Fig. 4d). We further confirmed that exogenous EGFP-VGLL3 formed nuclear puncta in NIH3T3 cells, whereas EGFP alone did not (Fig. 4e). These results led us to hypothesise that VGLL3 can undergo phase separation in the nucleus. To determine whether VGLL3-positive puncta have liquid-like properties, fluorescence recovery after photobleaching (FRAP) experiments were conducted in NIH3T3 cells expressing EGFP-VGLL3. After photobleaching, the signal for EGFP disappeared, and EGFP-VGLL3 puncta recovered, reaching ~60% of the pre-bleaching intensity in ~30 s (Fig. 4f, Supplementary Movie 1). These kinetics are similar to those of other reported liquid-like condensates[46–48]. To further characterise the VGLL3-positive puncta, we treated NIH3T3 cells expressing EGFP-VGLL3 with 1,6-hexanediol (1,6-HD), an aliphatic alcohol that is known to disrupt phase-separated condensates[49]. Upon 1,6-HD treatment, VGLL3-positive puncta in the cells rapidly disappeared (Fig. 4g), whereas treatment with 2,5-hexanediol (2,5-HD), an aliphatic alcohol which does not disrupt phase-separated condensates[49], did not affect the formation of VGLL3-positive puncta (Fig. 4g). These results indicate that VGLL3-positive puncta are liquid droplet-like and highly dynamic and that VGLL3 can phase separate into condensates in the nucleus.

### The intrinsically disordered region in VGLL3 is essential for its phase separation and promotion of collagen production

Intrinsically disordered regions (IDRs) exist in amino acid (aa) sequences of many proteins that undergo LLPS[47]. Using IUPred2A[50], we identified a high score region (aa 63–78), which might be the IDR responsible for the phase-separation of VGLL3 (Fig. 4h). Of the 16 amino acids in this region, 15 were acidic amino acids (13 glutamic acids and 2 aspartic acids). To confirm the contribution of this IDR to the phase separation of VGLL3, we generated a VGLL3 mutant that lacks the poly-glutamate IDR (aa 63–78), VGLL3 ΔIDR mutant and a VGLL3 mutant with its IDR domain changed to a glycine-and-serine-rich sequence, VGLL3 GGS mutant (Fig. 4h). Overexpression of VGLL3 ΔIDR mutant or VGLL3 GGS mutant in NIH3T3 cells, showed that the percentage of cells with puncta in the nucleus was dramatically decreased, compared with that in VGLL3 wild-type (WT) cells, indicating that the IDR is necessary for the phase separation of VGLL3 in the nucleus (Fig. 4i).

We further examined whether the ability of VGLL3 to undergo LLPS is required for the increased collagen expression. VGLL3 WT overexpression in NIH3T3 cells significantly increased the mRNA expression of *Col1a1* in the cells (Fig. 4j). However, *Col1a1* mRNA expression was not increased in NIH3T3 cells overexpressing the VGLL3 ΔIDR mutant or the VGLL3 GGS mutant, even though the protein levels of VGLL3 WT and the VGLL3 mutants were comparable (Fig. 4j). These results demonstrated that the increased expression of collagens by VGLL3 depends on its ability to undergo LLPS.

### VGLL3 is incorporated into non-paraspeckle NONO condensates

As shown in Fig. 4d, endogenous VGLL3 co-localised with endogenous NONO, a key component of paraspeckles in myofibroblasts. Close inspection of mass spectrometry analysis also revealed that about half (15/40) of the paraspeckle proteins are VGLL3 interactors (Supplementary Fig. 7), suggesting that VGLL3 is incorporated into paraspeckles[17,51]. To examine this, we detected the expression of

nuclear paraspeckle assembly transcript 1 (*Neat1*), a long noncoding RNA that functions as an essential framework of paraspeckles[17,22], in cardiac myofibroblasts by fluorescence in situ hybridisation (FISH). The combination of FISH and immunofluorescence staining for VGLL3 demonstrated that the number of VGLL3 puncta was much higher than that of *Neat1*. Most of the VGLL3-positive puncta were *Neat1*-negative, and only a few VGLL3-positive puncta were *Neat1*-positive (Fig. 5a, Supplementary Fig. 8a). Consistent with this data, most of the EGFP-VGLL3 puncta in NIH3T3 cells were *Neat1*-negative (Fig. 5b, Supplementary Fig. 8b). The co-staining for NONO and *Neat1* revealed that paraspeckles expressing both NONO and *Neat1* were present in cardiac myofibroblasts and the number of NONO puncta was much higher than that of *Neat1* (Fig. 5c), like other cells in previous reports[18,19,52], indicating the existence of non-paraspeckle NONO condensates in myofibroblasts. Very recent reports revealed that non-paraspeckle NONO condensates do not contain *Neat1* but are composed of many proteins that make up paraspeckles[18,19]. Consistent with these previous reports, we found that the non-paraspeckle NONO condensates in cardiac myofibroblasts contain splicing factor proline- and glutamine-rich (SFPQ), an RNA-binding protein[18,19,22] (Fig. 5d). Co-localisation of VGLL3 with NONO or SFPQ was also confirmed in NIH3T3 cells expressing small amounts of EGFP-VGLL3 (Fig. 5e, f). To further characterise the VGLL3/NONO-positive puncta, we treated myofibroblasts with RNase A. The RNA digestion almost completely eliminated nuclear VGLL3-positive puncta in cardiac myofibroblasts (Fig. 5g), suggesting that the formation of the VGLL3/NONO condensates depends on architectural RNA distinct from *Neat1*. Although *Neat1* was detected only in a small portion of VGLL3/NONO condensates (Supplementary Fig. 8a), *Neat1* knockdown significantly reduced collagen production in cardiac myofibroblasts (Supplementary Fig. 8c), suggesting that *Neat1* is also involved in fibrosis-related gene expression.

### Identification of EWSR1 as a VGLL3-interacting protein that promotes collagen expression in non-paraspeckle NONO condensates

To determine the protein involved in the VGLL3-mediated fibrotic pathway, we knocked down *Nono*, *Rbm14* and *Tardbp* encoding TDP-43 in cardiac myofibroblasts. However, their knockdown failed to decrease collagen expression (Supplementary Fig. 9a–c). We thus knocked down some RNA-binding proteins that are candidate VGLL3-interacting proteins (Supplementary Fig. 7) and found that *Ewsr1* knockdown decreased the expression of collagens (Fig. 5h, Supplementary Fig. 10a), demonstrating that EWSR1 is involved in increase of collagen mRNA expression in cardiac myofibroblasts. The decreased mRNA expression of collagens by *Vgll3* knockdown was the same as that with the co-knockdown of *Vgll3* and *Ewsr1* (Fig. 5i), suggesting that these participate in the same pro-fibrotic signalling pathway. We then confirmed the interaction between VGLL3 and EWSR1 in HEK293 cells overexpressing HA-VGLL3 and FLAG-EWSR1 using immunoprecipitation (Fig. 5j). To further reveal the endogenous interaction between VGLL3 and EWSR1 in cardiac myofibroblasts, we immunoprecipitated VGLL3 using a custom-made antibody in cardiac myofibroblasts treated with control siRNA or *Vgll3* siRNA (Fig. 5k). The protein amount of EWSR1 in cardiac myofibroblasts was not affected by *Vgll3* siRNA treatment (Fig. 5k). Parallel reaction monitoring (PRM) analysis, an MS/MS-based targeted quantification method, revealed that the amount of EWSR1 in the anti-VGLL3 immunoprecipitates was much higher than that in the control immunoprecipitates (Fig. 5k, Supplementary data 2). Further, the amount of EWSR1 peptides in the anti-VGLL3 immunoprecipitates was decreased upon *Vgll3* siRNA treatment (Fig. 5k, Supplementary data 2). VGLL3 peptides were also detected in the immunoprecipitates obtained from cardiac myofibroblasts with the VGLL3 antibody, but not in those from *Vgll3* siRNA-treated cardiac myofibroblasts or those obtained with the control antibody (Fig. 5k, Supplementary data 2), confirming that VGLL3 in the cells was

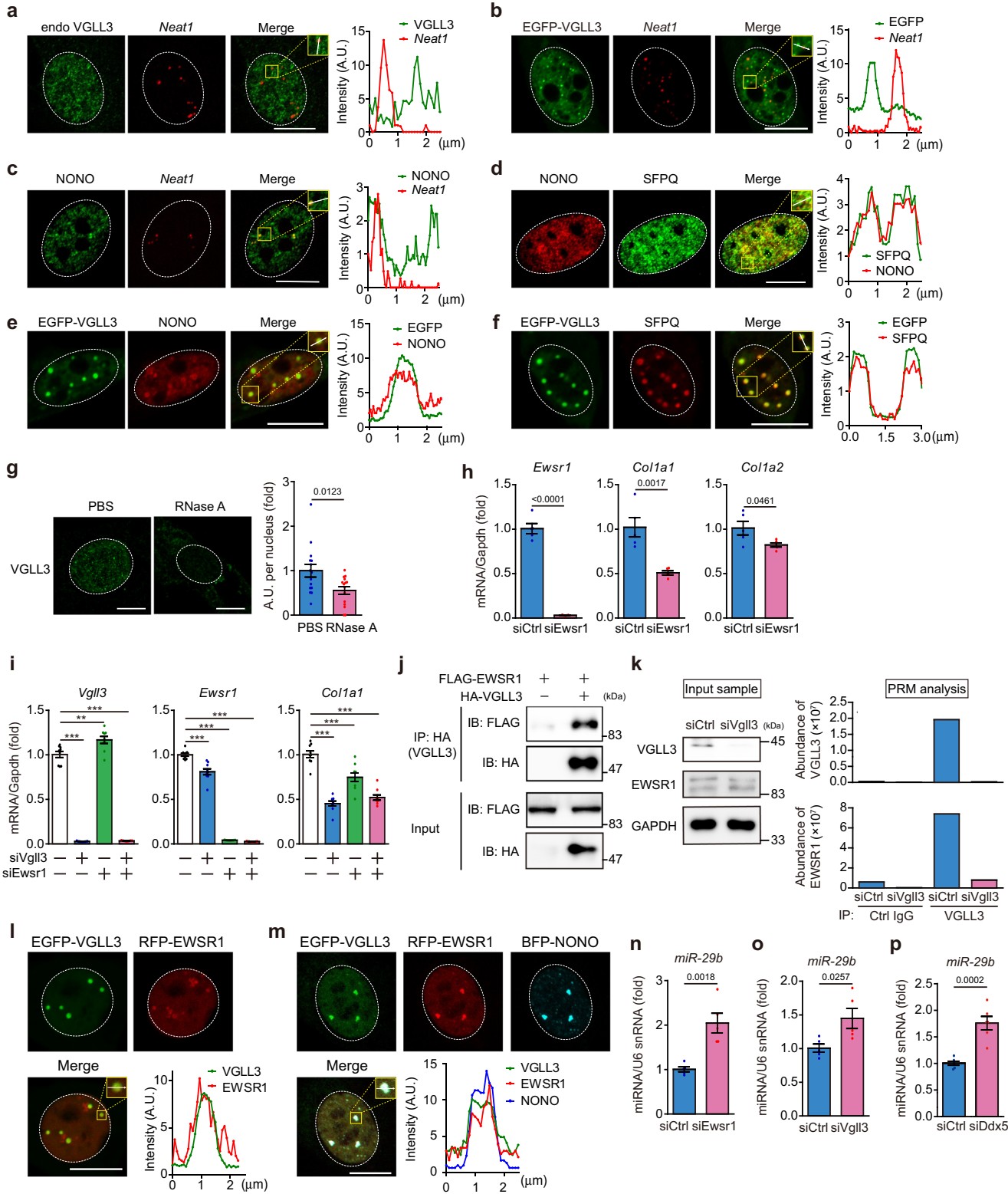

specifically immunoprecipitated with the anti-VGLL3 antibody. These data demonstrate the endogenous interaction between VGLL3 and EWSR1. Consistently, co-localisation experiments using NIH3T3 cells expressing EGFP-VGLL3 and RFP-EWSR1 revealed that the localisation of VGLL3 nuclear puncta and that of EWSR1 nuclear puncta was almost identical in NIH3T3 cells (Fig. 5l). However, the EGFP-VGLL3 ΔIDR puncta were localised in the cytoplasm and not co-located with RFP-EWSR1 puncta (Supplementary Fig. 10b). We further

confirmed that the localisation of these EGFP-VGLL3/RFP-EWSR1 puncta and BFP-NONO nuclear puncta was nearly identical in NIH3T3 cells (Fig. 5m).

## EWSR1 and VGLL3 decrease the miR-29b levels in cardiac myofibroblasts

NONO heterodimerises with SFPQ to process pri-miRNA prior to presentation to the Drosha complex[20]. In addition, EWSR1 deficiency was

**Fig. 5 | The VGLL3/EWSR1 complex in NONO condensates increases collagen expression through attenuation of miR-29b production. a** Micrographs of endogenous VGLL3 puncta detected by immunostaining, and *Neat1* mRNA detected by RNA-FISH in cardiac myofibroblasts. **b** Micrographs of EGFP-VGLL3 puncta and *Neat1* mRNA detected by RNA-FISH in NIH3T3 cells. **c** Micrographs of endogenous NONO puncta detected by immunostaining, and *Neat1* mRNA detected by RNA-FISH in myofibroblasts. **d** Micrographs of endogenous NONO and SFPQ detected by immunostaining in myofibroblasts. **e, f** Micrographs of EGFP-VGLL3 and endogenous NONO (**e**) or SFPQ (**f**) in NIH3T3 cells. **g** Micrographs of endogenous VGLL3 in myofibroblasts treated with PBS or RNase A. Summed intensity of VGLL3-signal per nucleus is shown in graph (*n* = 15 cells/group). **h** mRNA levels of *Ewsr1* and fibrosis-related genes in cardiac myofibroblasts transfected with siRNA targeting *Ewsr1* (*n* = 5 each). **i** *Vgll3*, *Ewsr1* and *Col1a1* mRNA levels in myofibroblasts transfected with siRNA targeting *Vgll3*, *Ewsr1*, or both *Vgll3/Ewsr1* (*n* = 5 each).

**j** Interaction between HA-VGLL3 and FLAG-EWSR1 in HEK293 cells. **k** Endogenous interaction between VGLL3 and EWSR1 in myofibroblasts analysed by parallel reaction monitoring (PRM) mass spectrometry of immunoprecipitants with anti-VGLL3 antibody. Western blot images of cell lysates used in the PRM analysis are shown in the left. **l, m** Images of live NIH3T3 cells transfected with EGFP-VGLL3 and RFP-EWSR1 (**l**) or EGFP-VGLL3, RFP-EWSR1, and BFP-NONO (**m**). **n–p** miR-29b levels in myofibroblasts transfected with siRNA targeting *Ewsr1* (**n**), *Vgll3* (**o**) or *Ddx5* (**p**) (*n* = 5–6). All experiments were performed at least three times. Data in (**g–i, n, o, p**) are shown as the mean ± SEM. *P*-values were determined using the two-sided Student's *t* test in (**g, h, n, o, p**), and one-way ANOVA followed by Tukey's range test in (**i**), \*\**P* < 0.01, \*\*\**P* < 0.001. White dashed circles mark the nucleus. The graphs represent line scans along the white lines in the yellow dashed square in the insets on (**a–f, l, m**). Scale bars in (**a–g, l**), and **m** = 10 μm. Source data are provided as a Source Data file.

reported to change the expression of three miRNAs (miR-18, miR-29b, and let-7f)[53], which suppress the expression of collagen and several fibrotic molecules[54–56]. We therefore speculated that the VGLL3/EWSR1 complex in NONO condensates controls the expression of collagens by controlling the biogenesis of these miRNAs. Of the three miRNAs whose expression levels are regulated by EWSR1, miR-29b levels were significantly increased in cardiac myofibroblasts after *Ewsr1* knockdown (Fig. 5n), while miR-18 and let-7f were not expressed in these cells. These results indicated that the expression of miR-29b[55], the best-known miRNA that suppresses the expression of collagen and several fibrotic molecules[54,55], most likely by binding to their 3′UTRs[57], and contributes to fibrosis after MI[58], is downregulated by *Ewsr1*. Importantly, miR-29b was significantly upregulated by *Vgll3* knockdown in cardiac myofibroblasts (Fig. 5o). We further examined the expression levels of miRNAs (miR-29a, miR-29c, miR-129 and miR-133a)[59], which have been reported to directly degrade collagen mRNAs, like miR-29b and other major miRNAs (miR-21 and miR-200a) that affect fibrotic responses in cardiac myofibroblasts[59]. Among these miRNAs, we found that *Vgll3* knockdown significantly increased miR-29a expression, and decreased miR-21 expression (Supplementary Fig. 11a). However, the expression of miR-29a and miR-21 was not significantly affected by *Ewsr1* knockdown (Supplementary Fig. 11b). These observations indicate that the main fibrosis-related miRNA whose expression was significantly affected by VGLL3/EWSR1 complex in NONO condensates in myofibroblasts is miR-29b.

Close inspection of the MS analysis that determined the candidate VGLL3-interacting proteins (Supplementary Fig. 12a) led us to find that endogenous VGLL3 interacts with endogenous DEAD-Box Helicase 5 (DDX5) (Supplementary Fig. 12b, Supplementary data 3), a component of microprocessor complex that is involved in microRNA biogenesis[60]. In fact, DDX5 was found in VGLL3/NONO condensates (Supplementary Fig. 12c). Importantly, *Ddx5* knockdown significantly increased miR-29b expression (Fig. 5p) and decreased the expression of collagens (Supplementary Fig. 12d), as did *Vgll3* knockdown, suggesting that DDX5 is involved in VGLL3-mediated suppression of miR-29b in the condensates.

## MI-induced cardiac fibrosis is attenuated in the absence of VGLL3

Finally, we investigated the contribution of VGLL3 and its suppression of miR-29b to fibrosis in vivo. We established *Vgll3* knockout (KO) mice (Fig. 6a, Supplementary Fig. 13a–c), and induced MI therein to elicit cardiac cell death and subsequent fibrosis. Because the decreased expression of miR-29b in the mouse heart exhibits a trough 3 days after MI surgery and increases soon thereafter[58], we decided to compare the expression levels of collagen mRNAs and miR-29b in the hearts of WT and *Vgll3* KO mice 3 days after MI. Three days after MI, the infarcted heart was divided into two parts: the infarcted section and remote (non-infarcted) section; the expression of fibrotic genes in each section was then determined. In WT control mice, *Col1a1*, *Col1a2*, *Col6a1*,

*Col14a1*, *Postn*, and *Vgll3* expression was remarkably increased in the infarcted area (Fig. 6a, Supplementary Fig. 14a). However, in *Vgll3* KO mice, the upregulation of these genes in the infarcted area was significantly attenuated (Fig. 6a, Supplementary Fig. 14a). Consistently, collagen production by myofibroblasts isolated from the infarcted hearts of *Vgll3* KO mice was significantly reduced compared to those isolated from the infarcted hearts of WT mice (Fig. 6b, Supplementary Fig. 14b). Both in vitro and in vivo proliferation assays demonstrated that the *Vgll3* KO and WT myofibroblasts do not differ in proliferative capacity (Fig. 6c, Supplementary Fig. 14c). In addition, the number of myofibroblasts at the infarcted area was unchanged between WT and KO mice (Supplementary Fig. 14d). Consistent with the results shown in Fig. 5n, miR-29b expression was significantly upregulated in the infarcted areas of *Vgll3* KO mice (Fig. 6d), suggesting that miR-29b expression is repressed by the VGLL3/EWSR1 axis in vivo. We then examined the effect of VGLL3 deficiency on the number and size of the non-paraspeckle NONO condensates in cardiac myofibroblasts. The immunocytochemical analysis using anti-NONO antibody demonstrated that the number and size of NONO condensates in myofibroblasts from KO mice were not significantly different from those in myofibroblasts from WT mice, although there was a trend towards a decrease (Supplementary Fig. 14e).

Comparison of cardiac conditions in WT and *Vgll3* KO mice at 28 days after MI using Picro-Sirius red staining of heart sections revealed a significantly lower degree of fibrosis in *Vgll3* KO mice than in WT mice (Fig. 6e, f). Furthermore, echocardiography demonstrated that the values for ejection fraction and fractional shortening rate, reflecting contractility, were significantly increased (Fig. 6g, Supplementary Fig. 14f, Supplementary Table 1) and E/A ratio, reflecting reduced diastolic function, was significantly decreased in *Vgll3* KO mice at 28 days after MI (Supplementary Fig. 14g). Together, these observations indicated that VGLL3 participates in fibrosis development in vivo.

## Discussion
In this study, we identified VGLL3 as a positive regulator for collagen production by myofibroblasts. *Vgll3* expression is induced immediately after MI, peaks on day 7 after MI, and remains high even on day 28 (Fig. 2a). During the acute inflammatory phase (day 0–day 3) after MI, ECM proteins are not highly accumulated. We thus consider that VGLL3 expression in this period is mainly induced by TGF-β that secreted by immune cells infiltrating the infarcted area. By the time acute inflammation subsides, the amount of TGF-β is decreased, but tissue fibrosis progresses, and the tissue becomes sclerotic. This increases the contribution of matrix stiffness to the induction of VGLL3 expression in myofibroblasts. In other words, the contribution of matrix stiffness to VGLL3 induction is expected to increase as the disease progresses into the chronic inflammatory phase.

Matrix stiffness induces VGLL3 nuclear translocation, resulting in increased expression of collagens and VGLL3, indicating the

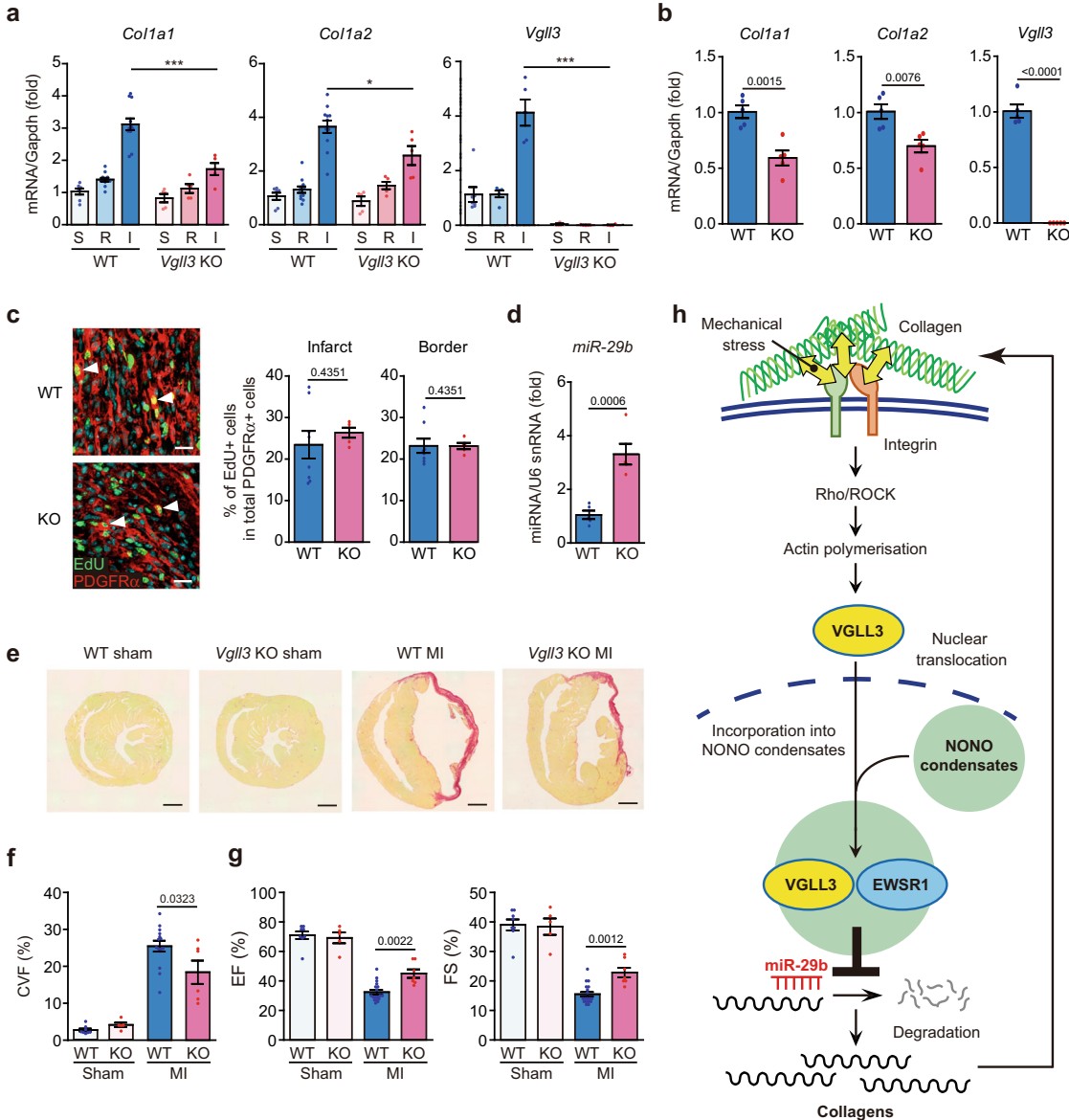

**Fig. 6 | *Vgll3* deficiency in mice attenuates cardiac fibrosis and impairs cardiac dysfunctions after myocardial infarction.** **a** mRNA levels of *Col1a1*, *Col1a2* and *Vgll3* in sham (S)-operated ventricles, and in the remote (R) and infarcted (I) areas of wild-type (WT) and *Vgll3* knock-out (KO) mouse hearts, 3 days after MI (WT: S/R/I, *n* = 7/12/12; KO: S/R/I, *n* = 5/5/5). **b** mRNA levels of *Col1a1*, *Col1a2* and *Vgll3* in cardiac myofibroblasts isolated from WT and *Vgll3* KO mouse hearts, 3 days after MI (WT: *n* = 5; KO: *n* = 5). **c** Immunostaining images of heart sections of WT and *Vgll3* KO mice on day 4 after MI. EdU were injected into the mice 24 h before sampling. White arrowheads indicate EdU+ and PDGFRα + fibroblasts. The percentages of EdU+ cells in total PDGFRα + fibroblasts of border or infarcted area are shown in the graph (WT: *n* = 8; KO: *n* = 5). **d** miR-29b levels in infarcted areas of WT and *Vgll3* KO mouse hearts, 3 days after MI (*n* = 5 each). **e** Images of heart sections of WT and *Vgll3* KO

mice, 28 days after MI, stained with Picro-Sirius red. **f** The collagen volume fraction (CVF) was calculated as the percentage of Picro-Sirius red-positive collagen deposition area (WT: Sham/MI, *n* = 8/14; KO: Sham/MI, *n* = 5/6). **g** Echocardiographic measurements of the ejection fraction (EF) and fractional shortening (FS), 28 days after MI (WT: Sham/MI, *n* = 8/18; KO: Sham/MI, *n* = 5/7). **h** Schematic of the VGLL3-mediated signalling pathway that enhances collagen gene expression in myofibroblasts. Data in (**a**–**d**, **f**, **g**) are shown as the mean ± SEM. *P*-values were determined using one-way ANOVA followed by Tukey's range test in (**a**) (*Col1a1* and *Col1a2*), Kruskal–Wallis followed by Dunn's test in (**a**) (*Vgll3*), two-sided Student's *t* test in (**b**, **d**, **f**), and two-sided Mann–Whitney's *U* test in (**c**, **g**), *\*P* < 0.05, \*\*\**P* < 0.001. n.s.; not statistically significant. Scale bars in **c** = 20 μm, **e** = 1 mm. Source data are provided as a Source Data file.

contribution of VGLL3 to the positive feedback loop of collagen production (Fig. 6h). Similar to VGLL3, the transcription cofactor MRTF also translocates to the nucleus upon mechanical stress, depending on the actin cytoskeleton, leading to an increase in ECM protein expression[13,61]. MRTF reportedly binds to G-actin under low mechanical stress. When G-actin assembles into F-actin in response to increased ECM stiffness, MRTF is released from G-actin, translocating to the nucleus, where MRTF binds to SRF. As for VGLL3, the upstream molecule that links actin polymerisation with VGLL3 nuclear translocation is yet unidentified. Elucidation of this molecule will further

enhance our understanding of the role of VGLL3 in fibrosis promotion by mechanical stress.

VGLL3 is a transcription cofactor proposed to be involved in some biological processes[25–27]. Among VGLL family proteins, only VGLL3 was found to promote fibrosis-related gene expression in myofibroblasts without TEAD1 (Supplementary Fig. 1f, g, 5d, 6). The amino acid sequences of other VGLL family molecules do not contain any domains similar to the glutamate-rich IDR domain of VGLL3. Thus, liquid–liquid phase separation by the glutamate-rich IDR domain, which is conserved in both mouse and human VGLL3, was found to play an

important role in the unique function of VGLL3. A previous report demonstrated that VGLL3 has a strong female-biased expression and is proposed to promote female-biased autoimmunity[27]. Thus, the LLPS of VGLL3 may also be involved in the development of autoimmune diseases and associated fibrosis. It would be of interest to analyse female *Vgll3* KO mice in detail from this perspective, though we have found no phenotypes in the *Vgll3* KO mice under normal conditions.

A recent report firstly demonstrated that substrate stiffness can influence the conditions of phase-separated condensates in cells[62]. They reveal that paraspeckles are mechanosensitive, and their parameters (i.g. number, area) are decreased by substrate stiffness in several cancer cell lines; however, the molecules responsible for the decrease have not been identified. In this study, we identified VGLL3 as a molecule that was induced by substrate stiffness and affects the miR-29b production in non-paraspeckle NONO condensates, where VGLL3 was incorporated, in cardiac myofibroblasts. These studies indicate that substrate stiffness may affect the conditions of various biomolecular condensates in a variety of cells and that the way in which it is affected may vary among different cell types. Our study revealed the existence of non-paraspeckle VGLL3/NONO condensates in cardiac myofibroblasts and its involvement in miR-29b production. However, our understanding of non-paraspeckle VGLL3/NONO condensates in the cells is just beginning. It will be important to identify architectural lncRNA other than *Neat1* and to unveil the detailed mechanism of miR-29b production in condensates in the future.

The VGLL3 ΔIDR mutant formed puncta in the cytoplasm, while VGLL3 WT was located in the nucleus. In addition to the low-complexity domain in the proteins undergoing LLPS, their post-translational modifications are also known to significantly affect their LLPS. TDP-43, a paraspeckle protein, was recently reported to be acetylated and phosphorylated by stimuli such as prolonged stress and to translocate outside the nucleus by the modifications[63,64]. In the cytoplasm, TDP-43 is incorporated into stress granules, and aggregated TDP-43 is proposed to contribute to the pathogenesis of amyotrophic lateral sclerosis[65]. Therefore, it will be interesting to examine the post-translational modifications of VGLL3 by matrix stiffness and their contribution to its nuclear translocation.

Fibrosis is involved in various diseases affecting multiple tissues, leading to the worsening of these conditions. However, pirfenidone (Esbriet®) and nintedanib (Ofev®) are the only approved anti-fibrotic drugs available at present. As such, novel therapies and target molecules are needed for this pathology. We found that VGLL3 promotes collagen production in both cardiac and hepatic myofibroblasts. Importantly, VGLL3 expression, which is normally low in the normal heart in mice and humans, is induced in fibroblasts in response to injury and mechanical signals. VGLL3 induction during myofibroblast differentiation was also observed in the fibrotic mouse liver. Collectively, VGLL3 is involved in the fibrosis of various organs and may represent a novel therapeutic target for tissue fibrosis.

## Methods
### Surgical procedure
All animal experiments were approved by the Animal Care and Use Committee of Kyushu University (A19-178-0, A21-139-0) and conform to the relevant national and international guidelines in the 'Act on Welfare and Management of Animals' (Ministry of Environment of Japan). WT mice (C57BL/6J and C57BL/6N) (Japan SLC Inc.) and *Vgll3* KO mice (C57BL/6N background) were used in this study. Only male mice were used for in vivo study, because female hormones have a variety of protective effects on the cardiovascular system. Mice were maintained under 12-h light/12-h dark cycle. The room temperature was regulated at 20 °C and humidity was controlled at 50%. C57BL/6J mice were used for the functional analysis of isolated myofibroblasts and identification of cells expressing VGLL3. C57BL/6N mice were used as the control group for experiments using *Vgll3* KO mice. Male and

female two-day-old SD rats were transferred from Kyudo and were immediately sacrificed.

For the mouse cardiac fibrosis model of MI, male mice (8–10 weeks old) were anaesthetised and subjected to permanent occlusion of the left coronary artery. Sham-operated animals underwent the same procedure without occlusion of the coronary artery.

For the model of $CCl_4$-induced liver fibrosis, male mice (8–10 weeks old) were intraperitoneally administered 5 mL/kg of 20% (v/v) $CCl_4$ solution [$CCl_4$ (05-1595-5; Sigma-Aldrich) was dissolved in corn oil (C8267; Sigma-Aldrich)] or vehicle (corn oil) twice weekly, for 4 weeks. To evaluate recovery from liver injury, $CCl_4$ administration was discontinued for 4 weeks after the final administration.

For the mouse model of NASH, male mice (6 weeks old) were fed a choline-deficient, L-amino acid-defined, high-fat diet with 60 kcal% fat and 0.1% methionine (CDAHFD) (A06071302N; Research Diets) for 10 weeks. Control mice were fed a standard diet (A06071314N; Research Diets) for 10 weeks.

### Preparation of CRISPR/Cas mRNA for generating *Vgll3* KO mice
pT7-sgRNA and pT7-hCas9 plasmid were kindly provided by Dr. Ikawa at Osaka University, Japan[66]. After digestion with EcoRI, hCas9 mRNA was synthesised using an in vitro RNA transcription kit (mMESSAGE mMACHINE™ T7 Ultra Transcription Kit; Ambion), according to the manufacturer's instructions. A pair of oligos targeting the genomic locus of exon 1 in *Vgll3* (NC_000082) were annealed and inserted into the BbsI site of the pT7-sgRNA vector. The sequences of the oligos were as follows: guide RNA 1; 5'- CGC TGC CAT GAG TTG TGC GG -3' and guide RNA 2; 5'- CCA GCC CGG CAG CCT TCC GG -3' (Supplementary Fig. 10a, b). After digestion with XbaI, gRNAs were synthesised using the MEGAshortscript™ Kit (Ambion). The precipitated RNA was dissolved in Opti-MEM I (Life Technologies) at 0.4 μg/μL.

### In vitro fertilisation (IVF) and electroporation
C57BL/6 N female mice (purchased from Crea-Japan Inc.) were used in this study. IVF was performed according to the protocol of the Center for Animal Resources and Development (at Kumamoto University, Japan) (http://card.medic.kumamoto-u.ac.jp/card/english/sigen/manual/onlinemanual.html). Electroporated embryos were cultured in KSOM medium and transferred on the next day to the oviducts of pseudo-pregnant females, on the day of the vaginal plug. A Genome Editor electroporator and LF501PT1-10 platinum plate electrode (BEXCo.Ltd.) were used for electroporation. In total, 50 embryos were prepared and subjected to electroporation. The collected embryos in KSOM medium were placed in the electrode gap filled with 5 μL of Opti-MEM I containing sgRNA and *hCas9* mRNA. Electroporation was performed 5 times at 25 V. The eggs were then cultured in KSOM medium at 37 °C and 5% $CO_2$ in an incubator until the two-cell stage.

### Genotyping
Genome editing of *Vgll3* was confirmed by genotyping (genotyping primer; *Vgll3*-S2: 5'-GCA GTA GCA GCT GGT TCA GC-3', *Vgll3*-AS2: 5'-CCC TGC CAT GCT CAG TTC TG-3') and genomic DNA sequencing (sequencing primer; *Vgll3*-S1: 5'-AAT GAA GGT GCC CGC GCA TG-3'); *Vgll3* mRNA in KO mice was also sequenced and the deficiency of WT *Vgll3* mRNA in KO mice was confirmed by qRT-PCR.

### Isolation of cardiac myofibroblasts using magnetic-activated cell sorting (MACS)
Three days after MI surgery, WT and *Vgll3* KO mice were sacrificed by an overdose of anaesthesia. After sacrifice, each mouse heart was collected in a 50 mL tube containing 10 mL of ice-cold phosphate-buffered saline (PBS, Nacalai Tesque) for storage. After the collection, the hearts were transferred to a 10 cm dish containing 5 mL ice-cold PBS on a clean bench. The infarcted parts of the hearts were cut into

small species by two surgical lancets and washed with ice-cold PBS. The heart pieces were transferred into a 50 mL tube, and 5 mL PBS containing 0.1% collagenase type II and 0.01% elastase (both from Worthington Biochemical Corporation) was added into the tube. The tube was incubated in a water bath at 37 °C with shaking for 10 min and the supernatant was collected using a 1000 μL pipette. After the collection, new 5 mL PBS containing 0.1% collagenase type II and 0.01% elastase was added into the tube and incubated at 37 °C with shaking for 10 min again. The collection of the supernatant and subsequent incubation was repeated five times. The supernatants were pooled and the pooled supernatants containing isolated cells were passed through a 70 μm EASYstrainer™ (Greiner) into a new 50 mL tube. The cell suspension was then centrifuged at 300 x g for 5 min and the supernatant was discarded after centrifugation. The cell pellet was suspended in 1 mL of red blood cell (RBC) lysis buffer (Roche) and incubated for 1 min at 20 °C. Following this, 9 mL of the culture medium was added to the cell suspension and centrifuged at 300 x g for 5 min. After removal of the supernatant, digested cardiac cells were suspended in culture medium [Dulbecco's Modified Eagle Medium (DMEM, Nacalai Tesque) containing 10% foetal bovine serum (FBS, Gibco) and 1% penicillin–streptomycin (Nacalai Tesque)] and allowed to attach to plastic culture plates overnight to remove debris. The attached cardiac cells were collected using Accutase (Nacalai Tesque). The collected cells were then treated with MACS buffer [0.5% bovine serum albumin (BSA, Sigma-Aldrich) and 2 mM EDTA in PBS] containing CD45 MicroBeads (1:5, 130-052-301; Miltenyi Biotec) for 20 min at 4 °C. After treatment, the cells were separated using an MS column and a MiniMACS separator (both from Miltenyi Biotec). CD45-positive cells were defined as macrophages and CD45-negative cells were defined as myofibroblasts.

To determine the purity of cardiac myofibroblasts, cells were stained with PE-conjugated anti-CD45 antibody (1:200, 30-F11; BioLegend), or FITC-conjugated anti-αSMA antibody (1:200, 1A4; Sigma-Aldrich) after permeabilisation with 0.5% saponin/PBS, and subsequently analysed using FACSVerse™ (BD).

### Regulation of myofibroblast differentiation using substrate stiffness

Cardiac myofibroblasts ($3 \times 10^5$ cells / 10 cm plate) or CCD-18Co cells ($1 \times 10^5$ cells / 10 cm plate) were cultured on plastic plates (Falcon) for 7 days in culture medium, and the total RNA or protein was obtained from the proliferated cardiac myofibroblasts ($1.6 \times 10^6$ cells / 10 cm plate) or CCD-18Co cells ($3.6 \times 10^5$ cells / 10 cm plate). A portion of the adherent cells was detached from the plate using 0.25% trypsin/1 mM EDTA solution (Nacalai Tesque) and the viability of the detached cells (viability of cardiac myofibroblasts: 96.1%, viability of CCD-18Co cells: 93.8%) was determined by Trypan blue staining (Wako). The cells were then transferred to an Ultra-low Attachment PrimeSurface® Cell Culture Plate (Diameter: 35 mm) (Sumitomo Bakelite) (cardiac myofibroblasts: $1.5 \times 10^6$ cells/ 35 mm plate, CCD-18Co cells: $3.4 \times 10^5$ cells/ 35 mm plate) and cultured in suspension (2 mL medium) for 7 days. Cells in suspension were collected by centrifugation, and the cell number (cardiac myofibroblasts: $1 \times 10^5$ cells/ 35 mm plate, CCD-18Co cells: $2.7 \times 10^4$ cells/35 mm plate) and viability (cardiac myofibroblasts: 100%, CCD-18Co cells: 91.7%) were measured by Trypan blue staining. Total RNA or protein was collected from the cells. The decrease in the cell numbers during suspension culture (cardiac myofibroblasts: $1.5 \times 10^6$ cells → $1 \times 10^5$ cells, CCD-18Co cells: $3.4 \times 10^5$ cells → $2.7 \times 10^4$ cells) was due to the anoikis, a form of programmed cell death that occurs in anchorage-dependent cells when they detach from the surrounding ECM. The cells in suspension culture were again cultured on a 10 cm plastic plate (cardiac myofibroblasts: $1 \times 10^5$ cells/10 cm plate, CCD-18Co cells: $2.5 \times 10^4$ cells/10 cm plate). After 7 days culture, the cell number (cardiac myofibroblasts: $1.5 \times 10^5$ cells/10 cm plate, CCD-18Co cells: $6.4 \times 10^4$ cells/10 cm

plate) and the viability (cardiac myofibroblasts: 86.7%, CCD-18Co cells: 89.6%) were measured by Trypan blue staining. Total RNA or protein was collected from the cells.

### Isolation of hepatocytes, Kupffer cells and hepatic stellate cells

The $CCl_4$-administered fibrotic liver was perfused with wash buffer [25 mM HEPES/0.5 mM EDTA/Hanks' balanced salt solution (HBSS, Nacalai Tesque)] warmed to 37 °C through the portal vein using a Perista® Pump (ATTO), followed by perfusion with digestion buffer [0.1% Collagenase A (Roche) /1.5 mM HEPES/HBSS]. The liver was then subdivided into small pieces, suspended in digestion buffer and shaken at 37 °C for 30 min. Centrifugation was performed twice at $50 \times g$ for 1 min to separate the precipitated cell fraction containing hepatocytes and the supernatant cell fraction containing Kupffer cells and HSCs. The precipitated cell fraction was plated on a 0.1% (w/v) gelatin-coated dish for 6 h to remove debris, and the adherent cells were collected using Accutase. The adherent cells were negative-sorted by MACS using CD45 MicroBeads; CD45-negative cells were defined as hepatocytes. The supernatant cell fraction was plated on a plastic dish and cultured overnight; the adhered cells were collected using Accutase and separated by MACS using F4/80 MicroBeads (1:10, 130-110-443; Miltenyi Biotec) and CD45 MicroBeads. F4/80-positive cells were defined as Kupffer cells and CD45-negative cells were defined as HSCs.

### Isolation of neonatal rat cardiac fibroblasts

Two-day-old SD rat (Kyudo) hearts were extracted and immediately digested in collagenase cocktail [PBS with 2 mg/mL Liberase™ research grade (Roche), 1 mg/mL fatty acid-free BSA (Sigma-Aldrich), and 1 mg/mL DL-β-hydroxybutyric acid (Sigma-Aldrich)]. The digested cells were suspended in culture medium (DMEM with 10% FBS and 1% penicillin–streptomycin), plated on plastic dishes, and incubated for 1 h at 37 °C. Adherent cells were then harvested as neonatal rat cardiac fibroblasts.

### Cell culture

HEK293T, NIH3T3 and CCD-18Co cell lines were purchased from American Type Culture Collection (ATCC). Plat-E cells were provided by Dr. Kitamura (Tokyo University, Japan). HEK293T, NIH3T3, Plat-E, cardiac myofibroblasts, HSCs, and neonatal rat cardiac fibroblasts were cultured in DMEM with 10% FBS and 1% penicillin–streptomycin. The culture medium of Plat-E cells was supplemented with 1 μg/mL puromycin (Sigma-Aldrich) and 10 μg/mL blasticidin S (Invitrogen). CCD-18Co cells were cultured in Minimum Essential Medium (Nacalai Tesque) with 10% FBS, 1 mM Sodium Pyruvate Solution (Nacalai Tesque), and 1% penicillin–streptomycin.

### Microarray analysis

Total RNA collected from the three above-mentioned cardiac myofibroblasts was assessed using SurePrint G3 Mouse Gene Expression 8x60K Microarrays (Agilent), followed by data analysis (TAKARA). The data were deposited in Gene Expression Omnibus (GEO) database under accession number GSE158236.

Hearts of mice that underwent myocardial infarction or sham surgery were rapidly frozen in liquid nitrogen. Total RNA from the hearts was extracted using ISOGEN (Nippon Gene), followed with purification using the RNeasy Mini Kit (QIAGEN). The purified RNA was assessed with the GeneChip Mouse Genome 430 2.0 Array (TORAY). The data were deposited in the GEO database under accession number GSE158157.

### Knockdown and overexpression analysis

Cardiac myofibroblasts, HSCs, and CCD-18Co cells were cultured on a Nunclon™ delta-treated plate (Nunc), and transfected with siRNA using Lipofectamine™ RNAiMAX (Invitrogen). Total RNA was collected 72 h after transfection. The siRNAs were purchased from Ambion (Supplementary Table 2).

For retroviral transduction of NIH3T3 cells, Plat-E cells were transfected with FLAG-VGLL3 (WT), FLAG-VGLL3 mutant (ΔIDR) or FLAG-VGLL3 mutant (GGS), as shown in Supplementary Table 3, using PEI-Max (24675; Polysciences). Forty-eight h after transfection, the supernatant containing the retrovirus was supplemented with 10 µg/mL polybrene (Sigma-Aldrich) and used for transducing NIH3T3 cells. Total RNA and protein were collected 96 h after the transduction.

### TGF-β stimulation

Neonatal rat cardiac fibroblasts were stimulated with 10 ng/mL of recombinant TGF-β (240-B; R&D Systems) for 24 h and differentiated to myofibroblasts.

### RNase A treatment

Cardiac myofibroblasts were washed with PBS and reaction buffer [20 mM Tris-HCl (pH7.5)/5 mM MgCl$_2$/0.5 mM EGTA/1×Protease Inhibitor] and permeabilized with 1% Tween 20 in reaction buffer for 10 min. The cells were then washed with reaction buffer and PBS and incubated with 100 µg/mL RNase A (R4642, Sigma-Aldrich) in nuclease buffer [5 mM MgCl$_2$/PBS], followed by immunocytochemistry.

### Plasmid construction

Each PCR product was ligated into pcDNA3 (Thermo) or pMXs-puro. To obtain full-length cDNAs, PCR was performed on cDNA synthesised from the mRNA of cardiac myofibroblasts and various mouse organs. The dsDNA of full-length VGLL3 carrying GGS mutation was synthesised by Integrated DNA Technologies and amplified by PCR. VGLL3 ΔIDR was constructed using the PCR-mediated plasmid DNA deletion method.

### Immunoprecipitation and western blotting

For transient transfection, HEK293T cells were transfected with plasmids with the pcDNA3 backbone, as shown in Supplementary Table 3, using X-tremeGENE 9 DNA Transfection Reagent (Roche). At 48 h after transfection, the cells were used for immunoprecipitation.

For retroviral transduction of cardiac myofibroblasts or NIH3T3 cells, Plat-E cells were transfected with plasmids with the pMXs-puro backbone, as shown in Supplementary Table 3, using X-tremeGENE 9 DNA Transfection Reagent. At 48 h after transfection, the supernatant containing the retrovirus was supplemented with 10 µg/mL polybrene and used for transducing the cardiac myofibroblasts or NIH3T3 cells. At 48 h after infection, cells were subjected to immunoprecipitation and immunoblotting.

For immunoprecipitation experiments using cells overexpressing exogenous VGLL3 and EWSR1, cells were lysed in RIPA buffer [50 mM Tris-HCl (pH 8.0)/150 mM NaCl/0.5% sodium deoxycholate/0.1% sodium dodecyl sulphate (SDS)/1% NP-40] containing protease inhibitors (1:100, Nacalai Tesque) and phosphatase inhibitors (1:100, Nacalai Tesque) at 4 °C for 20 min. The cell lysate supernatants then reacted overnight at 4 °C with the following magnetic beads-conjugated antibodies: anti-DYKDDDDK (FLAG) magnetic beads (1E6; Wako) and anti-HA magnetic beads (2–2.2.14; Thermo). The beads were then washed seven times with RIPA buffer. To elute proteins, the beads were boiled at 95 °C for 5 min with 2× SDS sample buffer [100 mM Tris-HCl (pH 6.8)/20% glycerol/2% SDS/0.04% bromophenol blue/2% 2-mercaptoethanol]. The supernatants of the cell lysates before the antibody reaction were mixed with 4× SDS sample buffer [200 mM Tris-HCl (pH 6.8)/40% glycerol/4% SDS/0.08% bromophenol blue/4% 2-mercaptoethanol] at a ratio of 3:1, boiled at 95 °C for 5 min, and used as input samples.

For immunoblotting to confirm the expression of exogenous VGLL3 and VGLL3 mutants, retrovirally transduced NIH3T3 cells were lysed in RIPA buffer containing protease inhibitors (1:100) and phosphatase inhibitors (1:100) at 4 °C for 20 min. The cell lysates were centrifuged, and the supernatants were mixed with 4× SDS sample buffer and boiled at 95 °C for 5 min.

For immunoblotting to detect Collagen I in cardiac myofibroblasts, NIH3T3 cells and CCD-18Co cells, the cells were lysed in RIPA buffer containing 1% 2-mercaptoethanol, protease inhibitors (1:100) and phosphatase inhibitors (1:100) at room temperature for 20 min and the lysates were boiled at 95 °C for 10 min. After centrifugation, the supernatants were collected and mixed with 4× SDS sample buffer.

Samples were separated on SDS-PAGE gels and transferred to PVDF membranes. The membranes were then incubated with the primary antibodies listed in Supplementary Table 4, followed by HRP-conjugated secondary antibodies. The bands were visualised using the ECL System (Perkin Elmer) and the intensity of the bands were quantified by Image Lab (Bio-rad).

### qRT-PCR analysis of mRNA

Cardiac myofibroblasts, NIH3T3 cells and CCD-18Co cells were lysed in RLT Plus buffer (QIAGEN) containing 1% 2-mercaptoethanol; cell lysates containing the total RNA were collected using QIAshredder (QIAGEN). Cardiac ventricles of mice that underwent MI or sham surgery were rapidly frozen in liquid nitrogen and homogenised in ISOGEN (Nippon Gene); the aqueous layer containing the total RNA was obtained using chloroform extraction. Total RNA was then purified using the RNeasy Mini Kit (QIAGEN). qRT-PCR was performed using the High-Capacity cDNA Reverse Transcription Kit (Life Technologies) followed by the TaqMan Fast Advanced Master Mix (Applied Biosystems), or Luna Universal qPCR Master Mix (New England BioLabs) on the StepOnePlus Real-Time PCR system (Applied Biosystems). The qRT-PCR probes were purchased from Applied Biosystems or Sigma-Aldrich (Supplementary Table 5), and *Gapdh* or 18 S rRNA was used as an internal control.

### qRT-PCR analysis of miRNA

Total RNA was purified from cardiac myofibroblasts using ISOGEN II (Nippon Gene) or miRNeasy Mini Kit (QIAGEN) or from mouse hearts using ISOGEN. miRNAs were reverse-transcribed using the TaqMan microRNA RT Kit (Applied Biosystems). qRT-PCR was performed using the Luna Universal qPCR Master Mix on the StepOnePlus Real-Time PCR system. TaqMan probes were purchased from Applied Biosystems (Supplementary Table 5) and U6 snRNA was used as an internal control.

### RNA-seq analysis

Library construction was performed by Macrogen Japan, using the TruSeq Stranded mRNA LT Sample Prep Kit (Illumina); the libraries were sequenced on the Illumina NovaSeq 6000. Sequence reads were analysed by FastQC and the adapter sequence, low quality bases and reads with length shorter than 36 bp were removed using Trimmomatic. Trimmed sequenced reads were mapped to the mouse reference genome (mm10) using HISAT2. Known genes and transcripts were assembled with StringTie; the read count was calculated and values were normalised as FPKM and TPM. The data were deposited in the GEO database under accession number GSE196194.

### Immunocytochemistry

The localisation of VGLL3, YAP1 and NONO was determined in cardiac myofibroblasts prepared from mouse hearts, three days after myocardial infarction. In some experiments, we retrovirally overexpressed FLAG-VGLL3 in the cells. Cells cultured in suspension in the Ultra-low Attachment PrimeSurface® Cell Culture Plate for 2 h were determined as the floating cell group. Myofibroblasts plated overnight on a poly-L-lysine (P4707; Sigma-Aldrich)-coated glass-bottom dish were determined as the attachment cell group.

In the experiment using hydrogel dishes, cells were cultured overnight on poly- L-lysine-coated Softview™ Easy Coat™ hydrogels bound to glass-bottom dishes (Matrigen) with different stiffnesses (1, 8, 25 and 50 kPa).

For the experiment that changed the adhesion time of the cells on the culture dish, cardiac myofibroblasts were cultured in suspension for 1 h, then seeded on poly-L-lysine-coated glass-bottom dishes for 0.5, 1, 2, and 4 h.

In the experiment using inhibitors, cardiac myofibroblasts were cultured overnight on a poly-L-lysine-coated glass-bottom dish, and then treated with the following inhibitors: 80 μM Y-27632 (Wako), 2 μM Latrunculin A (Abcam), 50 μM VS-4718 (ChemieTek), 30 μM BTT-3033 (TOCRIS), 3 μg/mL Cell Permeable C3 transferase (Cytoskeleton), and 50 μM Blebbistatin (Wako). VS-4718 was treated for 24 h and the other inhibitors were treated for 4 h.

For immunocytochemistry, cardiac myofibroblasts cultured under the above conditions were fixed with 1% paraformaldehyde (PFA)/PBS and stained with CellTrace™ Far Red (6 μM; Thermo) to label the entire cell. For permeabilisation of the cell membrane, 0.5% saponin/PBS was used in the experiment to determine the localisation of VGLL3 or YAP1 in floating or adherent cells, and 0.1% Triton X-100 was used in the other experiments. After blocking with 5% BSA/PBS for 1 h, cells were stained overnight at 4 °C with the following primary antibodies: rabbit anti-VGLL3 antibody (1:200, ab83555; Abcam), mouse anti-YAP1 antibody (1:200, WH0010413M1; Sigma-Aldrich), mouse anti-NONO antibody (1:200, RN013MW; MBL), rat anti-DYKDDDDK antibody (1:100, L-5; Novus). To visualise these antigens, cells were stained with the following species-specific secondary antibodies: AlexaFluor®488-conjugated donkey anti-rabbit IgG (1:200, Invitrogen), AlexaFluor® 546-conjugated goat anti-mouse IgG (1:200, Invitrogen), and AlexaFluor® 488-conjugated donkey anti-rat IgG (1:200, Invitrogen). After immunostaining, the nuclei were stained with DAPI (4′,6-diamino-2-phenylindole) solution (1:1000, Dojindo). Staining was evaluated under a confocal microscope. Images were acquired from randomly selected fields using a confocal microscope (LSM700; Carl Zeiss). To evaluate the degree of VGLL3 and YAP1 nuclear localisation, the average intensities of VGLL3 and YAP1 in the nuclei or cytoplasm were quantified using ImageJ (NIH), and the ratios of nuclear to cytoplasmic VGLL3 or YAP1 intensity were calculated. Staining to measure the number and size of NONO condensates in cardiac myofibroblasts was observed under a stimulated emission depletion microscope (TCS SP8 STED; Leica), and the size of NONO condensates was quantified by ImageJ.

## Immunohistochemistry

Mouse hearts were fixed overnight in 4% PFA/0.1 M phosphate buffer (PB), embedded in Tissue-Tek O.C.T. Compound (Sakura), and rapidly frozen in liquid nitrogen. Tissues were sectioned (6-μm thickness) using a cryostat (Cryostar NX70; Thermo Scientific), air-dried and permeabilised with 0.1% Triton X-100/PBS. After blocking with 5% BSA/PBS for 1 h, the sections were stained overnight at 4 °C, with the primary antibodies listed in Supplementary Table 4. To visualise these antigens, tissues were stained with the species-specific secondary antibodies listed in Supplementary Table 4. Staining was evaluated under a confocal microscope (LSM700). Quantitative analysis for cell numbers in heart sections was performed using ImageJ.

## In situ hybridisation

To detect *Vgll3* mRNA in MI-operated mouse hearts, Tissue-Tek O.C.T. Compound-embedded sections (6-μm thickness) of heart tissue fixed in 4% PFA/0.1 M PB were subjected to the RNAScope® Multiplex Fluorescent v2 Assay (ACD) according to the manufacturer's instructions. After *Vgll3* mRNA was labelled with tyramide-Cy5 (Perkin Elmer) and *Postn* mRNA was labelled with tyramide-Cy3 (Perkin Elmer), heart sections were blocked with 10% BSA/PBS and subjected to immunostaining.

To detect *VGLL3* mRNA in the autopsy heart specimens of patients who underwent myocardial infarction or to detect *Vgll3* mRNA in the liver sections of NASH model mice, paraffin-embedded tissue sections were deparaffinised and subjected to RNAScope® 2.5 HD–Red Assay

(ACD) according to the manufacturer's instructions. After *VGLL3* mRNA was labelled with Fast Red dye, the sections were blocked with 10% normal donkey serum/1% BSA/PBS and subjected to immunostaining.

To visualise various proteins, the mRNA-labelled samples described above were subjected to immunostaining with their primary antibodies, followed by the species-specific secondary antibodies; these antibodies are listed in Supplementary Table 4. After immunostaining, the nuclei were stained with DAPI (1:500 or 1:1000). Staining was evaluated under a confocal microscope (LSM700).

## Stellaris™ RNA FISH

Cardiac myofibroblasts or NIH3T3 cells transfected with EGFP-VGLL3 were labelled with mouse *Neat1* FISH probe with Quasar® 570 Dye (SMF-3010-1; LGC Biosearch Technologies) according to the manufacturer's sequential immunostaining methods. The antibodies used for immunostaining are listed in Supplementary Table 4. After immunostaining, the nuclei were stained with DAPI (1:1000). Staining was evaluated under a confocal microscope (LSM700).

## In vivo assessment for the intracellular localisation of EGFP-VGLL3 in cardiac myofibroblasts of fibrotic area

Plat-E cells were transfected with EGFP/pMXs-puro or EGFP-VGLL3/pMXs-puro using PEI-Max. Forty-eight h after transfection, the supernatant containing the retrovirus was concentrated by centrifugation (8000 × *g*) at 4 °C for 16 h, and precipitated retrovirus was suspended with PBS containing 4 μg/mL polybrene. For in vivo retroviral transduction of VGLL3 to cardiac myofibroblasts, this retroviral suspension (20 μL/heart) was administrated intramyocardially just below the ligation site immediately after the ligation of the coronary artery, using 29-gauge needles (BD Biosciences). Three days after administration, mouse hearts were fixed in 4% PFA/0.1 M PB for 3.5 h. The intracellular localisation of EGFP-VGLL3 or EGFP in cardiac myofibroblasts was assessed using immunohistochemistry.

## Picrosirius red staining

Mouse hearts were fixed in 10% formalin neutral buffer solution (Wako) for 24 h and embedded in paraffin. Then, 5-μm heart sections were deparaffinised and incubated in Picrosirius red buffer (3% Picric acid/0.15% Direct Red) for 1 h. Finally, heart sections were incubated in 0.01 N HCl for 1 min twice and subsequently subjected to dehydration. The stained sections were observed under a microscope (BZ-X700; Keyence) and analysed using a BZ-II analyzer (Keyence).

## Immunoprecipitation-mass spectrometry

Cardiac myofibroblasts retrovirally transduced with FLAG-VGLL3 or control vector were fixed with 0.1% PFA and lysed in a HEPES-RIPA buffer (20 mM HEPES-NaOH [pH 7.5]/1 mM EGTA/1 mM MgCl$_2$/150 mM NaCl/0.25% sodium deoxycholate/0.05% SDS/1% NP-40) containing protease inhibitors (1:100), phosphatase inhibitors (1:100) and 28 U/μL of Benzonase (1:500, Novagen) at 4 °C for 20 min. The supernatant of the cell lysate was incubated overnight at 4 °C with anti-FLAG magnetic beads. The beads were washed three times with HEPES-RIPA buffer and twice with 50 mM ammonium bicarbonate. Proteins on the beads were digested with 200 ng of Trypsin/Lysyl Endopeptidase mixture (Promega), and the digested peptides were reduced, alkylated, acidified and desalted using GL-Tip SDB (GL Sciences). The eluates were evaporated in a SpeedVac concentrator and dissolved in 0.1% trifluoroacetic acid and 3% acetonitrile (ACN). LC-MS/MS analysis of the resultant peptides was performed on an EASY-nLC 1200 ultra-high-performance liquid chromatograph connected to a Q Exactive Plus mass spectrometer through a nanoelectrospray ion source (Thermo Fisher Scientific). The peptides were separated on a 75 μm inner diameter × 150 mm C18 reversed-phase column (Nikkyo Technos) with a linear 4–32% ACN gradient for 0–150 min followed by an increase to 80% ACN for 20 min and finally hold at 80% ACN for 10 min. The mass

spectrometer was operated in a data-dependent acquisition mode with a top 10 MS/MS method. MS1 spectra were measured at a resolution of 70,000, an automatic gain control (AGC) target of $1 \times 10^6$, and a mass range from 350 to 1500 $m/z$. MS/MS spectra were triggered at a resolution of 17,500, an AGC target of $5 \times 10^4$, an isolation window of 2.0 $m/z$, a maximum injection time of 60 ms, and a normalised collision energy of 27. Dynamic exclusion was set to 15 s. Raw data were directly analysed against the SwissProt database, restricted to *Mus musculus*, using Proteome Discoverer version 2.4 (Thermo Fisher Scientific) with the Sequest_HT search engine for identification and label-free precursor ion quantification. The search parameters were as follows: (i) trypsin as an enzyme with up to two missed cleavages; (ii) precursor mass tolerance of 10 ppm; (iii) fragment mass tolerance of 0.02 Da; (iv) carbamidomethylation of cysteine as a fixed modification; and (v) acetylation of the protein N-terminus and oxidation of methionine as variable modifications. Peptides and proteins were filtered at a false-discovery rate (FDR) of 1% using the percolator node and the protein FDR validator node, respectively. Normalisation was performed such that the total sum of abundance values for each sample across all peptides was the same.

Data were analysed using a two-tailed Student's *t* test and the *p*-values were calculated. To control the FDR, calculated *p*-values were corrected using the Benjamini–Hochberg method and *q*-values (FDR-adjusted *p*-values) were calculated[67]. A volcano plot was used to determine the mean fold change and *q*-value of each FLAG-VGLL3-binding protein.

### Gene ontology enrichment analysis
The genes whose expression decreased upon si*Vgll3* treatment (M < −0.5, A > 0) or the proteins that interact with FLAG-VGLL3 in cardiac myofibroblasts (fold change >2, *q*-value <0.1) were analysed using DAVID Bioinformatics Resources version 6.8.

### Immunoprecipitation followed by PRM analysis
For immunoprecipitation to detect the endogenous interaction between VGLL3 and EWSR1, cardiac myofibroblasts were fixed with 0.1% PFA and lysed in IP buffer [20 mM HEPESNaOH (pH 7.5)/1 mM EGTA/1 mM MgCl$_2$/150 mM NaCl/5% glycerol/1% NP-40] containing protease inhibitors (1:100) and phosphatase inhibitors (1:100) at 4 °C for 20 min. The supernatants of cell lysates were incubated for 2 h at 4 °C with SureBeads Protein G (Bio-Rad) coupled with the custom-made rabbit polyclonal anti-VGLL3 antibody, raised against the amino acid sequence of mouse VGLL3 from position 307 to 325. The beads were then washed four times with IP buffer and twice with 50 mM ammonium bicarbonate. Proteins on the beads were digested with 200 ng of Trypsin/Lysyl Endopeptidase mixture, and the digested peptides were reduced, alkylated, acidified, desalted and dissolved in 0.1% trifluoroacetic acid and 3% ACN. To quantify VGLL3 and EWSR1, four peptides of VGLL3 and five peptides of EWSR1 were measured by PRM, an MS/MS-based targeted quantification method using high-resolution MS (Supplementary data 2). Targeted MS/MS scans were acquired by a time-scheduled inclusion list at a resolution of 70,000, an AGC target of $2 \times 10^5$, an isolation window of 2.0 $m/z$, a maximum injection time of 1 s, and a normalised collision energy of 27. Time alignment and relative quantification of transitions were performed using Skyline software[68].

### Live cell imaging and fluorescence recovery after photobleaching (FRAP)
For imaging the puncta in living cells, NIH3T3 cells were transfected with EGFP-VGLL3, RFP-EWSR1, RFP-DDX5 and/or BFP-NONO using PEI-Max. Then 24 or 48 h after the transfection, the cells were observed under a confocal microscope (LSM700; Carl Zeiss).

In the experiment using 1,6-HD and 2,5-HD, NIH3T3 cells overexpressing EGFP-VGLL3 were treated with 2.5% of 1,6-HD and 2,5-HD for 10 min.

The FRAP assay was performed using a confocal microscope (LSM700; Carl Zeiss). NIH3T3 cells overexpressing EGFP-VGLL3 WT were bleached by using a 488 nm laser at 100% laser power (150 consecutive irradiation). Images were then captured every 1 s using a time-lapse system for 90 s after photobleaching. To measure FRAP recovery, the intensity of puncta (I(t)) and the background (B(t)) were measured using Zeiss software. The intensity of pre-bleaching (I(pre)-B(pre)) was set to 100% and we calculated FRAP recovery according to the following formula: (the FRAP recovery (%) = (I(t) - B(t) / I(pre) - B(pre)) × 100).

### Identification of the intrinsically disordered region (IDR) by IUPred2A
The IDR scores of VGLL3 WT were calculated using the IUPred2A (https://iupred2a.elte.hu) for each amino acid. Each amino acid in VGLL3 WT was coloured using the PLAAC algorithm (http://plaac.wi.mit.edu).

### PCR analysis for the mRNA expression of VGLL family members
The mRNA expression of VGLL family members in cardiac myofibroblasts was evaluated by PCR using specific primer sets (Supplementary Table 6) and KOD-FX (Toyobo).

### Absolute quantification of mRNA
Absolute levels of mRNAs (*Vgll1-4*) in cardiac myofibroblasts were calculated using the standard curve method. The cDNA of cardiac myofibroblasts, FLAG-VGLL1/pcDNA3, FLAG-VGLL2/pcDNA3, HA-VGLL3/pMXs-puro and HA-VGLL4/pMXs-puro ($10^{-2}$ to $10^{-9}$ μg/μL) was measured by qRT-PCR using the probes for *Vgll1*, *Vgll2*, *Vgll3*, and *Vgll4*. The copy numbers/ng of total RNA of *Vgll1*, *Vgll2*, *Vgll3* and *Vgll4* were then determined.

### Echocardiography
Echocardiography was analysed from the short-axis view of the Left ventricular (LV) acquired by two-dimensional targeted M-mode tracings using a Nemio XG image-analysis system (SSA-580A; Toshiba Medical Systems). Left ventricular internal diameter (LVID) was measured as the largest anteroposterior diameter in either diastole (LVIDd) or systole (LVIDs). The fractional shortening (FS) was calculated as FS = [(left ventricular internal diameter in diastole (LVIDd) – left ventricular internal diameter in systole (LVIDs))/LVIDd] × 100. The ejection fraction (EF) was calculated as EF = [(end-diastolic volume (EDV) – end-systolic volume (ESV))/EDV] × 100. EDV and ESV were calculated according to Teichholz's method.

To assess LV diastolic function, pulsed wave Doppler imaging was performed using Prospect T1 (S-Sharp Corporation). LV filling peak velocity (E, mm/s) and atrial contraction flow peak velocity (A, mm/s) were acquired from the images of mitral valve Doppler flow from apical four-chamber view. The E/A ratio, a marker of the diastolic function of the left ventricle, was calculated.

All echocardiography measurements were performed under inhalation anaesthesia (1.5-2.0% of isoflurane at flow rate 2 L/min) to keep the heart rate between 450 and 520 bpm.

### Cardiac and hepatic *VGLL3* and fibrosis-related gene expression in patients with heart failure and NASH
The gene expression levels of *VGLL3* and fibrosis-related genes in the hearts of control individuals or patients with heart failure and in the livers of control individuals or patients with NASH were validated using data from respective GEO datasets (GSE116250: 14 controls and 50 patients with heart failure[41]; GSE126848: 14 controls and 16 patients with NASH[45]). A normalised count-based differential expression analysis was performed.

### EdU assay
Male mice were intraperitoneally administered with EdU (10 μg/g body weight) at 3 days after MI surgery. Then 24 h after EdU injections,

hearts were fixed in 4% PFA/0.1 M PB for 3.5 h. Before immunohistochemical staining, EdU in heart sections was detected using the Click-iT™ EdU imaging kits (C10337; Thermo Fisher Scientific) according to the manufacturer's instructions.

## WST-8 proliferation assay

Cell proliferation was determined by WST-8 assay. Myofibroblasts isolated from mouse hearts 3 days after MI were seeded on a 96-well plate at a density of $2.5 \times 10^3$ cells/well and transfected with siRNA. After transfection for 0, 24, 72 and 120 h, the culture supernatants were replaced with fresh culture medium containing 10% Cell Count Reagent SF (07553-15; Nacalai Tesque), and the cells were incubated at 37 °C in 5% $CO_2$ incubator. After 2 h incubation, the absorbance at 450 nm was measured using a microplate reader (EnSpire; Perkin Elmer).

## Statistics analysis

Statistical analyses were performed using GraphPad Prism 9.3.1. Data represent the means ± SEM of at least three independent experiments. We performed the Shapiro–Wilk test that determine the distribution of the variables for all data. Based on the respective Shapiro–Wilk test results, all data were analysed using parametric or nonparametric methods. Statistical significance was determined using the two-tailed unpaired Student's $t$-test (parametric) or Mann–Whitney $U$ test (nonparametric) for comparisons between two groups. For multiple comparisons, we used ANOVA (parametric) followed by the Tukey or Kruskal–Wallis tests (nonparametric), followed by the Dunn's test. Differences were considered statistically significant at $P < 0.05$.

## Study approval

Studies using human autopsy specimens were performed with the informed consent of participants, following the ethical guidelines of Jichi Medical University. The study protocol was approved by the ethics committee of Kyushu University (29–400).

## Reporting summary

Further information on experimental design is available in the Nature Portfolio Reporting Summary linked to this Article.

## Data availability

We declare that all the data supporting the findings of this study are available in the paper and its supplementary information files. The data of the microarray and RNA-seq analyses performed in this study have been deposited in the Gene Expression Omnibus under accession codes GSE158236 (microarray for Fig. 1d) and GSE158157 (microarray for Fig. 1d), and GSE196194 (RNA-seq for Fig. 3a). The MS proteomics data in this study have been deposited to the ProteomeXchange Consortium via the jPOST partner repository with the dataset identifiers PXD039091 (Fig. 4a), PXD039092 (Fig. 5k), and PXD039093 (Supplementary Fig. 12b). Source data are provided with this paper.

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

## Acknowledgements

The authors thank Dr. H. Kurose, Mr. K. Takayama and Ms. M. Nishida for contribution to the early stages of this work, and Ms. M. Kawano and Dr. K. Nishino for technical assistance. This study was supported by grants from Grants-in-Aid for Scientific Research (KAKENHI) [to M.N. (JP20H03383, JP17H03984, JP17H05510) and A.N. (JP19K07122)]; from The Takeda Science Foundation, The Mochida Memorial Foundation for

Medical and Pharmaceutical Research, The Salt Science Research Foundation, MSD Life Science Foundation, Senri Life Science Foundation, 2018 Bristol-Myers Squibb KK Research Grants, the Suzuken Memorial Foundation, SENSHIN Medical Research Foundation, the Nakatomi Foundation, the Koyanagi Foundation, Princess Takamatsu Cancer Research Fund, Ono Medical Research Foundation, Hoansha Foundation (to M.N.); from Japan Agency for Medical Research and Development (AMED) (JP17fk0210113h, JP20gm5810030h, JP21am0401003s0303 to M.N.); from Platform Project for Supporting Drug Discovery and Life Science Research [Basis for Supporting Innovative Drug Discovery and Life Science Research (BINDS)] from AMED under Grant Number JP19am0101091; from Joint Usage and Joint Research Programs of the Institute of Advanced Medical Sciences, Tokushima University (to M.N.); from Grant-in-Aid for JSPS Research Fellow (16J03451 to S.M., 19J20083 to Y.H., 21J11273 to T.H., 22J21002 to K.K., 22J20790 to N.T.); from JST SPRING, Grant Number JPMJSP2136. We appreciate for the technical support from the Research Support Center, Graduate School of Medical Sciences, Kyushu University, from Medical Institute of Bioregulation, Kyushu University, and from the Center for Advanced Instrumental and Educational Supports, Faculty of Agriculture, Kyushu University.

## Author contributions

M.N. designed research, performed experiments, and analysed and interpreted data. Y.H., S.M., C.T., T.M., T.H., K.K., Y.Y., N.T., and A.N. performed some in vivo and in vitro experiments and interpreted data. Y.H., S.M., C.T., and R.Y. performed animal surgery. H.K. performed interactome analysis. M.O. generated *Vgll3* KO mice. S.T. conducted microarray analysis. T.N. and A.T. analysed human samples. M.N. wrote the paper.

## Competing interests

The authors declare no competing interests.
