## [Peer Review File · Nature Communications]

REVIEWER COMMENTS

Reviewer #1 (Remarks to the Author):

This is a very well designed and executed study. The report is clearly detailed and easy to follow for the reader. The data are robust and connected such that the mechanism is probed nicely.

The significance of this work identifies VGLL3 as a regulator of collagen production by myofibroblasts, and subsequent fibrosis, in the context of heart and liver fibrosis. The in vitro and in vivo work is concordant.

The only comment that I have relates to figure 6 and why collagen is quantified at 3 days post-MI when this is likely too early to see the maximum differences? It may be that the VGLL3 KO mice showed no difference at later timepoints, but if collagens, Postn, and miR-29b were quantified (or are available for quantification) at later time-points (i.e. day 28 corresponding to figs 6c-e), that would be interesting to show. At the least, it should be clarified why 3 days post-MI was chosen as this is early in the fibrotic process when many inflammatory cells are invading and the myofibroblast population may not be dominant till later (i.e. 7 days +).

Reviewer #2 (Remarks to the Author):

The manuscript by Horii et al addresses an area of high relevant interest, relating mechanobiology and regulation of gene expression, and applying the knowledge in a medically relevant model.

This review focusses mostly on the exploration of the relationship of VGLL3 with paraspeckle proteins, paraspeckles and NEAT1.

In this reviewer's opinion the case for VGLL3 being found in paraspeckles has not been satisfactorily established, and is based on a number of non-definitive results.

A general issue in the manuscript is the heavy reliance on overexpression of tagged proteins for evaluation. The biophysical behaviour of proteins is highly concentration dependent and over-expression of proteins in the cell nucleus can easily result in artefacts. Many proteins, when overexpressed in the nucleus can form phase separated droplets which are not physiologically relevant. These droplets can behave as shown herein (FRAP recovery and hexanediol susceptible). The statement on p22-112 following the description of hexanediol treatment "These results indicate that VGLL3 is incorporated into paraspeckles." is simply not true.

The gold standard for paraspeckle localisation is colocalisation of endogenous protein (or where necessary, relatively lowly overexpressed protein) with NEAT1 (Stellaris FISH probes for NEAT1 are available and well-characterised), as well as NONO. It is essential that NEAT1 colocalisation is demonstrated. A VGLL3 antibody was used in Fig 1 - could this not be used for endogenous protein colocalisation studies?

The fluorescent micrographs in Figs 4 and 5, showing colocalisation of VGLL3 with NONO or EWSR1 are not convincing. The NONO (and to a lesser extent the EWSR1) puncta do not look like normal paraspeckles. A challenge with overexpression artefacts is that endogenous phase-separating proteins can be drawn from their native location to the artefactual aggregates.

While it is quite possible that the poly-glutamate IDR that has been identified in VGLL3 is involved in its apparent ability to phase separate, the choice of E->A mutagenesis to investigate it is probably unfortunate. Firstly, a minor detail, IUPred analysis of a heavily mutated protein is not a valid approach to deciding that its intrinsic disorder is reduced - this statement can simply be removed. Next, also minor, it is not clear why E but not D were mutated as they are chemically almost identical in this context. Most significantly, introduction of a long tract of alanine residues is almost guaranteed to have some kind of dominant effect. This is not a benign mutation. A more illuminating choice would have been to either delete the region, or replace it with glycine-and-serine-rich sequence. It does not surprise me at all that this protein precipitates into solid aggregates in the cytoplasm, but this

observation is more like a "null" rather than a "LLPS-knockdown".

Thus, while regulatory links between VGLL3, NEAT1, EWSR1 and miR-29b have been demonstrated, the evidence that its control occurs via paraspeckles or phase-separation is not yet convincing.

The dissolution of VGLL3 puncta by 1,6-hexanediol is not diagnostic of paraspeckle localisation, as stated. This is a generic (and rather crude) diagnostic for any liquid phase-separated body.

Relevant literature has not been addressed, for example Todorovski et al <https://doi.org/10.1091/mbc.E20-02-0097> which relates ECM, mechanical stiffness and NEAT1/Paraspeckles.

These comments focus on a criticism of the LLPS aspect of the manuscript, but it is of credit to the authors that they provide significant amounts of diverse information and experimental results on which to base that criticism. The criticism is not of the quality of the experiments, but some of the choices that have been made (tags, overexpression, mutation), which can in principle be addressed. If VGLL3 can be robustly demonstrated to localise to paraspeckles, then many interesting strands may be drawn together. As it stands the evidence is not robust.

Reviewer #3 (Remarks to the Author):

The mechanisms of how a cardiac myofibroblast controls its ECM production is of great interest, as it is associated with pathogenesis of cardiac remodeling. In this present study, Horii et al. identified Vgll3 as a gene upregulated by mechanical stimuli. They observed its translocation into nuclei on the stiff substrate. They also suggested that an essential domain of Vgll3 undergoes a liquid-liquid phase separation (LLPS), leading to incorporation into paraspeckles contributing to microRNA stabilization. Their experiments demonstrated that Vgll3 interacted with paraspeckle-related proteins including Ewsr1. In addition, they investigated phenotypes of Vgll3 knockout mice regarding collagen expression and systolic functions after myocardial infarction.

The authors performed well-designed in vitro experiments to investigate the intracellular kinetics of Vgll3 such as translocation into nuclei and LLPS. The results describing the binding proteins and the potential roles in paraspeckles were also convincing. However, the provided data of in vivo experiments were minimal and there is still a gap to fill between findings in vitro and those in vivo.

Major comments

1. The authors provide interesting data regarding the correlation of nuclear Vgll3 and collagen expression that appears to be contact dependent. This is a novel finding. A great deal of data is generated, but it is unclear how these are connected to the in vivo function of Vgll3.
2. The abstract is somewhat misleading. This study has limited data regarding "controlling myofibroblast differentiation". The authors should try to be specific and discuss collagen production rather than fibroblast differentiation/activation. Another example is page 9 line 15 where the authors are only examining collagen, α SMA, and VGLL3 transcript expression.
3. The authors use an unconventional method of cell separation and identification. They mention a previous publication, but that publication does not describe the phenomenon of dedifferentiation. Therefore, the authors need to give the audience more context on what is being achieved in their non-adherent cell population. What state does this represent? What are "ultra-low attachment plates"? Culture conditions should be defined. What density; how many cells die etc...? What is the profile of cells in the absence of MI or on the initial date of plating.
4. The term mechanical stimulation is a general term and not exact for the phenomenon that they are studying. A more descriptive term should be used. Possibly something related with attachment/substrate stiffness?
5. As mentioned in the Introduction, Vgll3 has been reported to enhance cell proliferation in some tumor cells. Proliferative activity of cardiac fibroblasts should be evaluated using both in vitro and in vivo assays. Especially, the number of (myo)fibroblasts and its proliferation in the hearts would be crucial factors to address anti-fibrotic outcomes of the Vgll3-KO mice.
6. The authors demonstrated that Vgll3 can interact with paraspeckle-related proteins. Then, does Vgll3 modulation affect the formation/structure? Evaluating appearance and the number of paraspeckles in Vgll3 deficiency would be required to show the impact of Vgll3 and its interactors on

paraspeckles.

7. For the in vivo studies, the authors need to assess the gene expression profiles not only in the whole heart lysate but in the isolated cardiac fibroblasts from infarcted hearts of Vgll3-KO mice to determine if these are consistent with the findings in vitro.
8. Eswr1 has been reported to regulate miR-29b via Drosha (Kim KY et al., Cell Death Differ. 2014). Also, this paraspeckle component protein regulate other micro RNAs targeting not only collagen but CTGF, known as one of the pro-fibrotic growth factors. Involvement of Drosha in the upregulation of miR-29b by Vgll3 deficiency should be examined. Moreover, it is not reasonable to conclude that the phenotype of Vgll3-KO is attributed to upregulation of miR-29b before evaluating CTGF. Conversely, does the interaction between Vgll3 and Eswr1 modulate other micro RNAs associated with fibrotic process? Please provide the data addressing these questions.
9. Provide representative images and detailed description of Method section for echocardiogram, e.g. dosage of anesthesia and view axis, because the values in the Fig. 6e looks far from the standard average in anesthetized mice described in a previous review (Lindsey M.L. et al. Am J Physiol Heart Circ Physiol. 2018). Additionally, diastolic functions and/or stiffness should be evaluated to demonstrate that Vgll3-KO results in functional alterations due to less collagen deposition in the scar.
10. It is unclear why the authors moved to in situ hybridization in vivo rather than immunohistochemistry when they clearly have an antibody that works. The images shown in figure 2 do not provide a good perspective on the extent of VGLL3 expression. Possibly, some quantification of the overall extent of overlap would be beneficial.
11. The authors should discuss if there are any phenotypes in the VGLL3 knockout.
12. The authors should not state that cardiac function was improved unless they have the temporal data to support this statement. Line 11 page 26
13. The authors need to justify that all statistical analyses were performed with parametric methods in this study. Otherwise non-parametric methods would be appropriate for the dataset without equality and normality.
14. The discussion has a great deal of results reiteration. Possibly, the authors can provide more of a summary rather than summarizing results a second time

Minor comments

1. It is suggested that the authors provide more current reviews in their citations.
2. The sentence in line 15, page 9 needs clarification. "Positive correlation" between differentiation by mechanical stress and Vgll3 expression level?
3. Remove an equal sign in line 14, page 13 and correct the sentence.
4. Avoid the use of "correlate" unless two continuous variables are analyzed statistically (line 10, page 8 etc.). Otherwise quantify "mechanical stimulus-dependent myofibroblast differentiation" and the expression level of Vgll3 followed by correlation analysis.
5. P 9 line 7 expression of this gene
6. No figure is referred to in the discussion of Rho/Rock signaling.
7. Can the authors provide better examples of nuclear stains in figure 1K?

Point-by-point response to the reviewers' comments

We thank the three reviewers for their insightful assessment of our study, which provided us with the opportunity to improve our manuscript. To address the reviewers' concerns, we have performed additional experiments and added a detailed description of our methods. The point-by-point responses to the reviewers' comments are provided below, and all revised text is denoted by red font in the manuscript.

Reviewer #1:

This is a very well designed and executed study. The report is clearly detailed and easy to follow for the reader. The data are robust and connected such that the mechanism is probed nicely.

The significance of this work identifies VGLL3 as a regulator of collagen production by myofibroblasts, and subsequent fibrosis, in the context of heart and liver fibrosis. The in vitro and in vivo work is concordant.

The only comment that I have relates to figure 6 and why collagen is quantified at 3 days post-MI when this is likely too early to see the maximum differences? It may be that the VGLL3 KO mice showed no difference at later timepoints, but if collagens, Postn, and miR-29b were quantified (or are available for quantification) at later time-points (i.e. day 28 corresponding to figs 6c-e), that would be interesting to show. At the least, it should be clarified why 3 days post-MI was chosen as this is early in the fibrotic process when many inflammatory cells are invading and the myofibroblast population may not be dominant till later (i.e. 7 days +).

Response: We appreciate the reviewer's favourable and valuable input. To explain our rationale behind choosing to conduct quantification after 3 days, we refer to a report stating that the decreased expression of miR-29b in the mouse heart exhibits a trough 3 days after MI surgery, increasing soon thereafter (Ref #58: Proc Natl Acad Sci U S A. 2008;105(35):13027-32). Based upon this information, we decided that the third day after MI surgery, representing the trough of miR-29b expression, is the appropriate time to observe the increase in miR-29b production due to VGLL3 deficiency. Appreciating that your query highlighted the lack of this information in our manuscript, we have included the aforementioned reason in the Results section of the revised manuscript on page 28, lines 2–5.

In addition, as per your suggestion to consider later timepoints, we measured the expression levels of fibrotic genes in mouse hearts on day 7 after MI surgery. We found that the mRNA levels of *Colla1* and *Colla2* were significantly decreased, and miR-29b expression levels were

significantly increased in the hearts of KO mice after MI, illustrated in Fig. R1:

Fig. R1 Vgll3 deficiency in mouse attenuates the expression of collagens and increases the miR-29b expression on day 7 after MI surgery.

(a, b) mRNA levels of the fibrosis-related genes (a) and miR-29b expression (b) in sham (S)-operated ventricles and in remote (R) and infarcted (I) areas of wild-type (WT) and *Vgll3* knock-out (KO) mouse hearts, 7 days after MI (WT: S/R/I, n = 5/6/6; KO: S/R/I, n = 4/4/4). Data are presented as the mean \pm SEM. *P* values were determined using Newman-Keuls multiple comparison tests (a) or unpaired two-tailed Student's *t*-tests (b), **P* < 0.05, ***P* < 0.01, ****P* < 0.001

Reviewer #2:

The manuscript by Horii et al addresses an area of high relevant interest, relating mechanobiology and regulation of gene expression, and applying the knowledge in a medically relevant model.

This review focusses mostly on the exploration of the relationship of VGLL3 with paraspeckle proteins, paraspeckles and NEAT1.

In this reviewer's opinion the case for VGLL3 being found in paraspeckles has not been satisfactorily established, and is based on a number of non-definitive results.

Response: We wish to express our sincere appreciation to the reviewer, who is a pioneer in the field of paraspeckle research, for insightful and valuable comments.

In the original manuscript, we judged condensates containing VGLL3 to be paraspeckles simply because they were positive for NONO, an essential structural component of paraspeckles. However, as you pointed out, to determine whether the condensates are paraspeckles, it is also essential to demonstrate their expression of *Neat1*, an architectural lncRNA essential for paraspeckle formation (Ref #22: McCluggage, F. & Fox, A. H. *Bioessays*. 2021;43: e2000245).

Accordingly, we conducted crucial *Neat1* RNA FISH staining (using the reagents kindly

indicated by you). Indeed, we found that most condensates containing VGLL3 were in fact non-paraspeckle NONO condensates (Neat1-negative), which has not been previously reported in myofibroblasts, although some VGLL3-positive condensates were paraspeckles (Neat1-positive). Based on this finding, we have rewritten the entire manuscript and performed several additional experiments (Co-localisation of VGLL3 and SFPQ in Fig. 5f, page 23, lines 8–10, Condensate dissolution following RNase A treatment in Fig. 5g, page 23, lines 10–13, Co-localization of VGLL3, NONO and EWSR1 in Fig. 5m, page 25, lines 15–17, Identification of DDX5, the component of microprocessor, as a molecule that binds to VGLL3 in Fig. 5p and Supplemental Fig. 12 page 27, lines 6–14) on the non-paraspeckle NONO condensates that contain VGLL3, describing the involvement of these condensates in the VGLL3-mediated fibrotic pathway.

We believe that the results of these experiments, following your suggestion, greatly improved the quality and content of our study.

A general issue in the manuscript is the heavy reliance on overexpression of tagged proteins for evaluation. The biophysical behaviour of proteins is highly concentration dependent and overexpression of proteins in the cell nucleus can easily result in artefacts. Many proteins, when overexpressed in the nucleus can form phase separated droplets which are not physiologically relevant. These droplets can behave as shown herein (FRAP recovery and hexanediol susceptible). The statement on p22-112 following the description of hexanediol treatment "These results indicate that VGLL3 is incorporated into paraspeckles." is simply not true.

Response: Thank you for providing valuable information that has not been found in the literature. In accordance, we amended the description "1,6-hexanediol (1,6-HD), an aliphatic alcohol that is known to disrupt paraspeckles" to read "1,6-hexanediol (1,6-HD), an aliphatic alcohol that is known to disrupt phase-separated condensates", on page 20, lines 9–10.

We have been careful in the interpretation of our experiments using overexpressed proteins for purposes of evaluation. To address your concern, we performed experiments that do not depend on the overexpression of tagged proteins whenever technically possible (Fig. 4d, 5a, Supplementary Fig.8a, 5c, 5d, and 5g). New data have been presented in the revised manuscript (Fig. 4d, 5a, Supplementary Fig.8a, 5c, 5d, and 5g) and the findings were reported on page 19, lines 15–17, page 22, line 10–16, page 22, line 18– page 23, line 4, page 23, lines 6–8, page 23, lines 11–13.

When it was unavoidable to use tagged proteins, the expression of tagged proteins in each cell was kept as low as possible by reducing the number of plasmids and the length of time after transfection. As a result, the number of puncta of the tagged protein (e.g. EGFP-VGLL3) in the nucleus was reduced. We performed most of the related experiments again and replaced the

data with new ones (Fig. 4e, 4f, 4g, 4i, 5b, 5e, 5f, 5l, and Supplementary Fig. 10b). The findings were reported on page 19, line 17–page 20, line 1, page 20, lines 2–6, page 20, lines 7–13, page 21, lines 10–14, page 22, lines 16–18, page 23, lines 8–10, page 25, lines 10–15.

The gold standard for paraspeckle localisation is colocalisation of endogenous protein (or where necessary, relatively lowly overexpressed protein) with NEAT1 (Stellaris FiSH probes for NEAT1 are available and well-characterised), as well as NONO. It is essential that NEAT1 colocalisation is demonstrated. A VGLL3 antibody was used in Fig 1 - could this not be used for endogenous protein colocalisation studies?

Response: We agree with your assessment and adjusted our approach accordingly. In addition to NONO localisation, we examined the localisation of NEAT1 in cardiac myofibroblasts using Stellaris FiSH probes, as per your suggestion. We identified paraspeckles in which both NEAT1 and NONO were localised in myofibroblasts. The new results are discussed under “VGLL3 is incorporated into non-paraspeckle NONO condensates”, as described on page 22, line 10–page 23, line 17, and illustrated in Fig. 5c. Importantly, the number of NEAT1 puncta was much lower than that of VGLL3 puncta, and only a few VGLL3 puncta merged with NEAT1 puncta (illustrated by Fig. 5a, 5b, and Supplementary Fig. 8a). Considering that VGLL3 co-localised with NONO (Fig. 4d, 5e), these new results indicated that VGLL3 resides mainly in non-paraspeckle NONO condensates, the existence of which has not been reported for cardiac myofibroblasts. We therefore appreciate your input and have revised the entire manuscript accordingly.

In our revision, we characterised the non-paraspeckle NONO condensates containing VGLL3 in cardiac myofibroblasts. Two recent papers reported non-paraspeckle NONO condensates with biological functions in other cells (Ref #18: Yasuhara T et al., Mol Cell 2022;82: 2738-2753; and Ref #19: Zhang S et al., bioRxiv preprint doi: <https://doi.org/10.1101/2022.02.28.482217>). Similar to these reports, we found that non-paraspeckle NONO condensates in cardiac myofibroblasts contained SFPQ, an RNA-binding protein with multiple biological functions including transcriptional regulation and miRNA processing (Fig. 5d). We further discovered that the formation of the VGLL3/NONO condensates in cardiac myofibroblasts was sensitive to RNase digestion, suggesting that RNA other than NEAT1 was essential for the formation of these condensates (Fig. 5g).

Importantly, we also found that endogenous VGLL3 interacts with DDX5, a component of the microprocessor complex that is involved in microRNA biogenesis, and reported this on page 27, lines 6–10, supported by Supplementary Fig. 12a and b. DDX5 was found in the VGLL3/NONO condensates (Supplementary Fig. 12c) and affected the expression of miR-29b

(Fig. 5p) and *Colla1* and *Colla2* (Supplementary Fig. 12d), supporting the notion that the condensates are involved in miR-29b production.

The VGLL3 antibody used in Fig. 1 can be used for endogenous protein co-localization studies. Using this antibody, we demonstrated that endogenous VGLL3 co-localised with endogenous NONO, as shown in Fig. 4d. In addition, we examined the co-localization of endogenous VGLL3 with Neat1 in cardiac myofibroblasts (Fig. 5a).

The fluorescent micrographs in Figs 4 and 5, showing colocalisation of VGLL3 with NONO or EWSR1 are not convincing. The NONO (and to a lesser extent the EWSR1) puncta do not look like normal paraspeckles. A challenge with overexpression artefacts is that endogenous phase-separating proteins can be drawn from their native location to the artefactual aggregates.

Response: As mentioned in a previous response, when it was unavoidable to utilize tagged proteins for evaluation purposes, we kept the expression of tagged proteins in each cell as low as possible by reducing the number of plasmids present and the length of time after transfection to avoid overexpression artefacts. We repeated most of the experiments which uses the expression of tagged proteins and replaced the data with new ones, amending Fig. 4e, 4f, 4g, 4i, 5b, 5e, 5f, 5l, and 5m). Essentially, we obtained the same results as before.

In the revised manuscript, we identified paraspeckles via co-staining with NONO and Neat1 in cardiac myofibroblasts, and the paraspeckles appeared normal (Fig. 5c). We also re-examined the co-localization of endogenous VGLL3 and endogenous NONO using immunostaining in the cells (Fig. 4d). This experiment, in tandem with the co-staining experiments between NEAT1 and endogenous VGLL3 (Fig. 5a), confirmed that VGLL3 is incorporated into non-paraspeckle NONO condensates rather than paraspeckles.

We also observed co-localization of VGLL3 and EWSR1 even when the number of nuclear VGLL3 puncta was reduced (Fig. 5l).

While it is quite possible that the poly-glutamate IDR that has been identified in VGLL3 is involved in its apparent ability to phase separate, the choice of E->A mutagenesis to investigate it is probably unfortunate. Firstly, a minor detail, IUPred analysis of a heavily mutated protein is not a valid approach to deciding that its intrinsic disorder is reduced - this statement can simply be removed. Next, also minor, it is not clear why E but not D were mutated as they are chemically almost identical in this context. Most significantly, introduction of a long tract of alanine residues is almost guaranteed to have some kind of dominant effect. This is not a benign mutation. A more illuminating choice would have been to either delete the region, or replace it with glycine-and-serine-rich sequence. It does not surprise me at all that this protein precipitates into solid

aggregates in the cytoplasm, but this observation is more like a "null" rather than a "LLPS-knockdown".

Response: Thank you for your constructive insights, based on which we implemented several changes. In the revised manuscript, we firstly removed the description and the related figure of IUPred analysis for VGLL3 mutant harbouring many mutations in the poly-glutamate IDR region. Next, we constructed two new VGLL3 mutants: (1) a VGLL3 mutant that lacks the poly glutamate IDR; and (2) a VGLL3 mutant with an IDR domain changed to a glycine-and-serine-rich sequence (not only E, but also D and K in the domain were mutated) (please refer to Fig. 4h). These mutants were expressed in NIH3T3 cells at very low levels and neither mutant formed bright puncta in the nucleus (Fig. 4i), indicating that the poly glutamate IDR region was essential for forming puncta in the nucleus. Moreover, we found that the VGLL3 mutants were unable to promote collagen expression (Fig. 4j). Kindly refer to revised text at page 21, line 6–page 22, line 2.

Using FRAP experiments, we found that puncta of EGFP-VGLL3 GGS mutant in cytoplasm hardly recovered after photobleaching, suggesting that the mutant forms puncta with low fluidity. However, in lieu of the comment that “It does not surprise me at all that this protein precipitates into solid aggregates in the cytoplasm, but this observation is more like a ‘null’ rather than a ‘LLPS-knockdown’”, we removed the results of FRAP experiments against VGLL3 puncta found in cytoplasm of NIH3T3 cells expressing the VGLL3 mutant.

Thus, while regulatory links between VGLL3, NEAT1, EWSR1 and miR-29b have been demonstrated, the evidence that is control occurs via paraspeckles or phase-separation is not yet convincing.

Response: We have performed a series of new experiments as per your suggestions to characterise the non-paraspeckle NONO condensates containing VGLL3 (as detailed in a previous response). We believe that the results of these experiments support our view that the phase separation of VGLL3 and its incorporation into non-paraspeckle NONO condensates (which have not been previously found in cardiac myofibroblasts) contribute to the VGLL3-mediated fibrotic pathway in cardiac myofibroblasts. We wish to express our appreciation to this reviewer for the insightful comments that led us to discover non-paraspeckle NONO condensates in myofibroblasts.

The dissolution of VGLL3 puncta by 1,6-hexanediol is not diagnostic of paraspeckle localisation, as stated. This is a generic (and rather crude) diagnostic for any liquid phase-separated body.

Response: Thank you for pointing this out. Based on your comment, we have modified the

description of the experiment using 1,6-hexanediol. Specifically, "1,6-hexanediol (1,6-HD), an aliphatic alcohol that is known to disrupt **paraspeckles**" was corrected to "1,6-hexanediol (1,6-HD), an aliphatic alcohol that is known to disrupt **phase-separated condensates**". (page 20, line 9–10).

Relevant literature has not been addressed, for example Todorovski et al <https://doi.org/10.1091/mbc.E20-02-0097> which relates ECM, mechanical stiffness and NEAT1/Paraspeckles.

Response: Thank you for highlighting this important paper, which demonstrates the involvement of phase-separated condensates in mechanobiology. We have cited it (Ref #62: Todorovski et al., MCB, 2020) as a reference in our revised manuscript and discussed pertinent content from this report on page 31, line 18–page 32, line 13.

These comments focus on a criticism of the LLPS aspect of the manuscript, but it is of credit to the authors that they provide significant amounts of diverse information and experimental results on which to base that criticism. The criticism is not of the quality of the experiments, but some of the choices that have been made (tags, overexpression, mutation), which can in principle be addressed. If VGLL3 can be robustly demonstrated to localise to paraspeckles, then many interesting strands may be drawn together. As it stands the evidence is not robust.

Response: Thank you again for your positive and professional feedback, which has guided us to conduct as many alternative experiments as possible. We believe that their results enhanced the quality and validity of our conclusions. It has led us to discover non-paraspeckle NONO condensates that were recently focused on other cells (Ref #18: Yasuhara T et al., Mol Cell 2022;82: 2738-2753; Ref #19: Zhang S et al., bioRxiv preprint doi: <https://doi.org/10.1101/2022.02.28.482217>) and to describe their involvement in the VGLL3-mediated fibrotic pathway.

We could ultimately conclude that non-paraspeckle NONO condensates (previously completely unknown in myofibroblasts) regulate myofibroblast collagen production in conjunction with the liquid-liquid phase separation of VGLL3, which senses mechanical stimuli. We hope that this finding will contribute to myofibroblast research and fibrosis studies.

Reviewer #3:

The mechanisms of how a cardiac myofibroblast controls its ECM production is of great interest, as it is associated with pathogenesis of cardiac remodeling. In this present study, Horii et al. identified Vgll3 as a gene upregulated by mechanical stimuli. They observed its translocation into nuclei on the stiff substrate. They also suggested that an essential domain of Vgll3 undergoes a liquid-liquid phase separation (LLPS), leading to incorporation into paraspeckles contributing to microRNA stabilization. Their experiments demonstrated that Vgll3 interacted with paraspeckle-related proteins including Ewsr1. In addition, they investigated phenotypes of Vgll3 knockout mice regarding collagen expression and systolic functions after myocardial infarction. The authors performed well-designed in vitro experiments to investigate the intracellular kinetics of Vgll3 such as translocation into nuclei and LLPS. The results describing the binding proteins and the potential roles in paraspeckles were also convincing. However, the provided data of in vivo experiments were minimal and there is still a gap to fill between findings in vitro and those in vivo.

Response: We would like to thank the reviewer for their valuable input. We have performed experiments (mainly *in vivo*) to address all concerns raised, as we shall highlight with responses to the comments below. We trust that this reviewer will understand that our paper mainly focuses on cellular biological analyses that demonstrate the stiffness-dependent nuclear translocation and induction of VGLL3 and the contribution of VGLL3 liquid-liquid phase separation to collagen production by myofibroblasts.

Major comments

1. The authors provide interesting data regarding the correlation of nuclear Vgll3 and collagen expression that appears to be contact dependent. This is a novel finding. A great deal of data is generated, but it is unclear how these are connected to the in vivo function of Vgll3.

Response: As you have mentioned, we found new and interesting insights into VGLL3 from a perspective of cell biology (mechanobiology and LLPS) and have thus summarised our findings in a cell biological paper. Nevertheless, we agree that it is important to demonstrate that VGLL3 is involved in fibrosis *in vivo*. For this reason, we generated *Vgll3* KO mice to perform essential *in vivo* experiments.

It is our understanding that this reviewer wishes for us to demonstrate the stiffness-dependent nuclear translocation of VGLL3 *in vivo*. We set out to demonstrate that VGLL3 resides predominantly in the nucleus of myofibroblasts in the fibrotic region, where increased stiffness exists. We first attempted to use immunohistochemistry with an anti-VGLL3 antibody. However,

we found that neither the commercially available VGLL3 antibody nor our custom-made VGLL3 antibody were applicable.

As a result, we administered a retrovirus carrying EGFP-VGLL3 intramyocardially to WT mice immediately following induction of MI by surgery. Kindly refer to our section titled “*In vivo* assessment for the intracellular localisation of EGFP-VGLL3 in cardiac myofibroblasts of fibrotic area”, as can be found on page 53, lines 3–14. Three days after MI surgery and subsequent retrovirus administration, EGFP-VGLL3 expression was detected in the fibrotic area of infarcted mouse hearts (Supplementary Fig. 2h). Furthermore, EGFP-VGLL3 expression was only observed in myofibroblasts in the infarcted area (Fig. 1k). We hypothesise that this may be because retroviruses infect only proliferating cells. Importantly, EGFP-VGLL3 was selectively localised in the nucleus only (Fig. 1k), whereas the EGFP control was uniformly present in both the nucleus and cytoplasm of myofibroblasts in the fibrotic area (Supplementary Fig. 2i). These results suggested that VGLL3 was translocated to the nucleus depending on the stiffness in the fibrotic areas.

We believe that these results contribute to filling the gap between our findings *in vitro* and *in vivo*.

2. *The abstract is somewhat misleading. This study has limited data regarding “controlling myofibroblast differentiation”. The authors should try to be specific and discuss collagen production rather than fibroblast differentiation/activation. Another example is page 9 line 15 where the authors are only examining collagen, αSMA, and VGLL3 transcript expression.*

Response: We agree with your observation. Accordingly, we have replaced the description “controlling myofibroblast differentiation” with “controlling collagen production by myofibroblasts” in the abstract (page 3, lines 3–4, 4–5 and 15). We have revised the sentence (Page 9, line 15 in our previous manuscript) as follows:

Previous manuscript: “Collectively, these observations indicate that *VGLL3* mRNA expression in myofibroblasts is positively correlated with **their differentiation state** regulated by mechanical stimuli.”

Revised manuscript: “Expression of *Vgll3* mRNA in myofibroblasts or *VGLL3* mRNA in CCD-18Co is positively correlated with **mRNA expression of collagen** regulated by substrate stiffness (Supplementary Fig. 1j)” (Page 9, line 16–Page 10, line 1)

In addition, we have changed all descriptions related to this point throughout the revised manuscript (page 8, lines 15–16; page 9, lines 12–13).

Previous manuscript: “*Vgll3* expression is positively correlated with mechanical stimulus-dependent myofibroblast **differentiation**”

Revised manuscript: “*Vgll3* expression is positively correlated with substrate stiffness-dependent **expression of collagen** by myofibroblasts” (Page 8, lines 16)

Previous manuscript: “We then examined whether *Vgll3* expression was affected by mechanically regulated **differentiation** of myofibroblasts from other organs and species”

Revised manuscript: “We then examined whether *Vgll3* expression was affected by substrate stiffness in myofibroblasts from other organs and species.” (Page 9, lines 12–13)

3. The authors use an unconventional method of cell separation and identification. They mention a previous publication, but that publication does not describe the phenomenon of dedifferentiation. Therefore, the authors need to give the audience more context on what is being achieved in their non-adherent cell population. What state does this represent? What are “ultra-low attachment plates”? Culture conditions should be defined. What density; how many cells die etc...? What is the profile of cells in the absence of MI or on the initial date of plating.

Response: Thank you for your constructive comments. The descriptions and citations about the cell separation of myofibroblasts and other cells in the previous manuscript may have been confusing and difficult to follow. To address this issue, we have rewritten the methodology for cell separation and identification to include more detail, both in the Result section (page 7, lines 12–18) and in the Methods section (page 37, line 2–page 38, line 15). Specifically, we have removed the citations and added the following description (page 7, lines 12–18):

“In brief, we digested the fibrotic mouse hearts using enzymes and removed the erythrocytes. Then, the constituent cells were subjected to overnight culture on plastic plates. After that, the attached cells were collected and subjected to magnetic-activated cell sorting (MACS) separation using anti-CD45 antibody. CD45-negative cells were collected as myofibroblasts. Almost all the collected cells were positive for α -smooth muscle actin (α SMA), a myofibroblast marker protein (Supplementary Fig. 1a, b).”

To clarify the conditions of myofibroblasts cultured in “ultra-low attachment plates”, we added the description “Culture in suspension” in red font in Fig. 1a. Ultra-low attachment plates are coated with hydrophilic substances which reduce cell adhesion greatly. Thus, when cardiac myofibroblasts cultured on plastic plates are transferred to these ultra-low attachment plates, they hardly attach to these plates and form aggregates. The deprivation of stiffness in their surrounds leads myofibroblasts to decrease their expression of collagens and the myofibroblast marker gene, *Acta2* (which encodes α SMA) (Fig. 1b, c). In contrast, the expression level of *Oct4*, a marker of undifferentiated cells, was increased in non-adherent myofibroblasts in a previous study

(Supplementary Figure 1e). These results suggest that myofibroblasts cultured in suspension (nonadherent myofibroblasts) are de-differentiated.

To further address your concerns, we expanded on the characterisation of the state of myofibroblasts that had been cultured in suspension for 7 days. We examined the expression levels of fibroblast marker genes (*Thy1* and *Tcf21*) (Ref #29: Ivey MJ, Tallquist MD. *Circ J*. 2016;80(11):2269-2276) or mesenchymal stem cell marker genes (*Islr* and *Nt5e*) (Ref #30: Hara A et al., *Circ Res*. 2019; 125: 414–430. #31: Takahashi M et al., *Front Cell Dev Biol*. 2021; 9: 749924.) in non-adherent myofibroblasts. The results demonstrated that the expression levels of *Islr* and *Nt5e* were significantly increased in non-adherent myofibroblasts, whereas those of *Thy1* and *Tcf21* were significantly decreased in non-adherent myofibroblasts (page 8, lines 4–7, and Supplementary Fig. 1e). These results suggest that the non-adherent cell population was de-differentiated and acquired mesenchymal stem cell-like properties.

Moreover, as per your suggestion, we defined the culture conditions (cell number, cell survival, and plate size) in detail and described them in the Methods section on page 38, line 17–page 40, line 5.

We are uncertain about the exact meaning of the reviewer's comment: “What is the profile of cells in the absence of MI or on the initial date of plating?” We only used cells isolated from mouse hearts after MI and did not isolate resident fibroblasts from healthy mouse hearts. Thus, we had no information regarding the cell profile in the absence of MI. However, we can present the profile of cells isolated from infarcted mouse hearts on the initial plating date. FACS analysis demonstrated that most of the cells isolated as cardiac myofibroblasts were CD45-negative and α SMA-positive (Supplementary Fig. 1a, b)

4. The term mechanical stimulation is a general term and not exact for the phenomenon that they are studying. A more descriptive term should be used. Possibly something related with attachment/substrate stiffness?

Response: In accordance with your comments, we replaced the term "mechanical stimulus" with "substrate stiffness" or "matrix stiffness" (depending on the context) as much as possible in the Introduction, Results, and Discussion sections, as well as in the Figure legends (page 3, line 8; page 6, line 5; page 7, lines 2, 8, and 10; page 8, line 15; page 9, lines 12, 15, and 18; page 10, lines 3 and 10; page 11, lines 7, 9, and 12; page 11, line18–page 12, line1; page 12, lines 4, 16–17; page 15, lines 15 and 18; page 16, lines 1, 5, 7, and 9; page 30, lines 9, 10, and 13; page 33, line 4; page 76, lines 2, 5, and 9; and page 87, line 15).

5. As mentioned in the Introduction, *Vgll3* has been reported to enhance cell proliferation in some tumor cells. Proliferative activity of cardiac fibroblasts should be evaluated using both *in vitro* and *in vivo* assays. Especially, the number of (myo)fibroblasts and its proliferation in the hearts would be crucial factors to address anti-fibrotic outcomes of the *Vgll3*-KO mice.

Response: We agree with your comments and have therefore evaluated the effect of VGLL3 on the proliferation of cardiac (myo)fibroblasts using both *in vitro* and *in vivo* assays. Our *in vitro* WST-8 assay is described in a section titled “WST-8 proliferation assay” on page 61, line 12–page 62, line 1, supported with Supplementary Fig. 14c. The *in vivo* EdU assay is described in a section titled “EdU assay” on page 61, lines 5–10, supported with Figure 6c. The results from these two tests demonstrated that VGLL3 did not significantly affect the proliferative activity of cardiac (myo)fibroblasts.

Furthermore, we counted the number of (myo)fibroblasts in the border and infarct areas of mouse hearts after MI induction. The numbers in WT mice were almost the same as those in the VGLL3 KO mice, as have been reported in Supplementary Fig. 14d.

6. The authors demonstrated that *Vgll3* can interact with paraspeckle-related proteins. Then, does *Vgll3* modulation affect the formation/structure? Evaluating appearance and the number of paraspeckles in *Vgll3* deficiency would be required to show the impact of *Vgll3* and its interactors on paraspeckles.

Response: Following constructive comments by reviewer #2, we adjusted our experimental approach. In addition to NONO localisation, we examined the localisation of NEAT1 in cardiac myofibroblasts using Stellaris FISH probes. We identified paraspeckles in which both NEAT1 and NONO were localised in myofibroblasts. All new results are discussed under “VGLL3 is incorporated into non-paraspeckle NONO condensates”, as described on page 22, line 5–page 23, line 17, and illustrated in Fig. 5c. Most importantly, the number of NEAT1 puncta was much lower than that of VGLL3 puncta, and only a few VGLL3 puncta merged with NEAT1 puncta (illustrated by Fig. 5a, 5b, and Supplementary Fig. 8a). Considering that VGLL3 co-localised with NONO (Fig. 4d, 5e), these new results indicated that VGLL3 in fact mainly resides in non-paraspeckle NONO condensates, rather than in paraspeckles.

Next, with your comment in mind, we evaluated the immunofluorescence of non-paraspeckle NONO condensates (instead of paraspeckles) in cardiac myofibroblasts isolated from the fibrotic hearts of WT and VGLL3 KO mice. The confocal microscopes used in our experiments had insufficient resolution to measure the number and size of endogenous non-paraspeckle NONO condensates correctly. Therefore, we employed a stimulated emission

depletion (STED) microscope. The number and size of the NONO condensates tended to decrease in myofibroblasts from VGLL3 KO mice, although the difference was not significant (Supplementary Figure 14e). We speculated that VGLL3 did not have such a large effect on the number and size of NONO condensates because VGLL3 is a modulator of NONO condensates, but not an essential component for the formation of condensates. The relevant revisions in our manuscript can be found on page 29, lines 2–7 (Results), and on page 50, lines 13–16 (Methods).

7. For the in vivo studies, the authors need to assess the gene expression profiles not only in the whole heart lysate but in the isolated cardiac fibroblasts from infarcted hearts of Vgll3-KO mice to determine if these are consistent with the findings in vitro.

Response: Once again, we agree with your observation and thank you for the input. Accordingly, we isolated cardiac myofibroblasts from infarcted hearts of WT or *Vgll3* KO mice and measured the expression levels of several fibrotic genes, such as *Colla1* and *Colla2*, in the cells. We found that expression levels of these genes were significantly decreased in cardiac myofibroblasts isolated from *Vgll3* KO mice (Fig. 6b), which is consistent with the data obtained using whole heart lysates (Fig. 6a). These revisions to our manuscript were included on page 28, lines 11–13.

8. Eswr1 has been reported to regulate miR-29b via Drosha (Kim KY et al., Cell Death Differ. 2014). Also, this paraspeckle component protein regulate other micro RNAs targeting not only collagen but CTGF, known as one of the pro-fibrotic growth factors. Involvement of Drosha in the upregulation of miR-29b by Vgll3 deficiency should be examined. Moreover, it is not reasonable to conclude that the phenotype of Vgll3-KO is attributed to upregulation of miR-29b before evaluating CTGF. Conversely, does the interaction between Vgll3 and Eswr1 modulate other micro RNAs associated with fibrotic process? Please provide the data addressing these questions.

Response: Indeed, the previous study you mention (Ref #53: Kim KY et al., Cell Death Differ. 2014) presented a schematic diagram (in their Fig. 6g, shown here on the bottom right) in which miR-29b appears to target both *Col4a1* and CTGF (encoded by *Ccn2*) in mouse embryonic fibroblasts. However, we have reread this paper carefully and conclude that it presents data that miR-29b targets *Col4a1* (in their Fig. 2e and 2f left, shown on the upper left of the next page), but it does not present data that miR-29b targets CTGF. Instead, it shows that miR-18b targets CTGF (in their Fig. 2e lower, 2f right, shown on the next page). In summary, they demonstrated that miR-29b targets *Col4a1* and miR-18b targets CTGF. We believe that the schematic diagram presented in their Fig. 6g was confusing and created a corrected scheme, based on their data, as

shown below.

Corrected scheme based on the data
(Fig. 2e 2f) in the Kim KY et al.,
 Cell Death Differ. 2014

In addition, it is well recognised that miR-29b targets the mRNA of various collagens, but it is less well known that *Ccn2* (which encodes CTGF) is a target gene of miR-29b.

Given these circumstances, we set out to confirm whether miR-29b targets *Ccn2* (which encodes CTGF) in cardiac myofibroblasts. Transfection of an miR-29b mimic reduced the expression of *Colla1* and *Colla2*, but not the amount of *Ccn2* in the cells (see Fig. R2 below). These results indicate that *Colla1* and *Colla2*, but not *Ccn2*, are target genes of miR-29b in cardiac myofibroblasts.

Fig. R2 The mimic of miR-29b suppressed the expression of *Colla1* and *Colla2* but not *Ccn2* in cardiac myofibroblasts.

mRNA levels of *Colla1*, *Colla2*, and *Ccn2* in cardiac myofibroblasts transfected with miR-29b mimic ($n = 5$ each). Data are presented as mean \pm SEM. P values were determined using an unpaired two-tailed Student's t-test, ** $P < 0.01$, *** $P < 0.001$.

Consistently, we found that *Vgll3* knockdown did not decrease the expression of *Ccn2* in cardiac myofibroblasts (Fig. R3). Thus, we concluded that *Ccn2* (encodes CTGF) is not a target gene of miR-29b, at least in cardiac myofibroblasts.

Fig. R3 Vgll3 knockdown did not decrease the expression of *Ccn2* in cardiac myofibroblasts. mRNA levels of *Ccn2* in cardiac myofibroblasts transfected with siRNA targeting *Vgll3* ($n = 5$ each). Data are presented as mean \pm SEM. P values were determined using unpaired two-tailed Student's t -test; * $P < 0.05$.

MiR-29b is the best-known and well-established miRNA that greatly contributes to mRNA production of various collagens (Ref #55: Kriegel, AJ et al., *Physiol. Genomics*. 2012). However, in accordance with the reviewer's comments, we searched for other miRNAs of which expression is regulated by both VGLL3 and EWSR1, which affect the mRNA production of various collagens.

Among the 21 miRNAs that exhibited a highly increased or decreased expression in EWSR1 KO mouse embryonic fibroblasts (Fig. 1a in Kim KY et al., *Cell Death Differ.* 2014), we found three miRNAs (miR-18, miR-29b, and let-7f) reportedly involved in the mRNA production of fibrotic molecules, including collagens (Ref #53: Kim KY et al., *Cell Death Differ.* 2014; Ref #55: Kriegel, AJ et al., *Physiol. Genomics*. 2012;80(11):237-244; Ref #56: Zhao et al., *Peer J*. 2022;10:e14097). As a result, we examined the involvement of miR-18b and let-7f in the VGLL3-mediated fibrotic pathway. We knocked down *Vgll3* and *Ewsr1* in cardiac myofibroblasts and measured the expression levels of miR-18b and let-7f. Unexpectedly, we found that neither miR-18b nor let-7f was expressed in the cells, indicating that both miRNAs were not relevant to the VGLL3-mediated fibrotic pathway, at least in cardiac myofibroblasts. We have described this in the revised manuscript on page 26, lines 3–10, under the heading “EWSR1 and VGLL3 decrease the miR-29b levels in cardiac myofibroblasts”.

Furthermore, we investigated the expression of miR-29a, miR-29c, miR-129, and miR-133a, all of which (like miR-29b) reportedly target the mRNA of collagen directly for degradation (Ref #59: O'Reilly, S et al., *Arthritis Res. Ther.* 2016;18:11). In contrast to miR-18b and let-7f, miR-29a, miR-29c, miR-129, and miR-133a were expressed in cardiac myofibroblasts. Among the miRNAs, only two miRNAs (miR-29a and miR-21) was significantly changed in siVGLL3-

knockdowned cells, suggesting that the other miRNAs were not relevant to the VGLL3-mediated fibrotic pathway (reported in Supplementary Fig. 11a).

We further examined whether the expression of miR-29a and miR-21 were influenced by EWSR1. The examination demonstrated that the expression was not significantly altered by EWSR1 knockdown, although the knockdown tended to increase miR-29a expression (Supplementary Fig. 11b). Therefore, we have focused on miR-29b in our manuscript. In summary, miR-29b is the only miRNA that targets collagen, and its expression is regulated by both VGLL3 and EWSR1.

We have described these data (Supplementary Fig. 11a, b) in the revised manuscript (page 26, line 14–page 27, line 5).

The previous study (Kim KY et al., *Cell Death Differ.* 2014) reported that *Ewsr1* deficiency increased the expression of Drosha, which in turn increased miR-29b expression. Thus, we examined whether *Vgll3* deficiency promoted miR-29b expression by increasing Drosha expression in cardiac myofibroblasts. However, *Vgll3* deficiency did not increase Drosha mRNA expression in the cells (Fig. R4).

Fig. R4 *Vgll3* deficiency did not increase the expression of *Drosha* in cardiac myofibroblasts. Drosha mRNA levels in cardiac myofibroblasts isolated from WT and *Vgll3* KO mice ($n = 5$ each). Data are presented as mean \pm SEM. The P value was determined using Mann-Whitney's U test, n.s.: not statistically significant.

We then searched for other molecules that could link VGLL3 and miR-29b expression levels. Our mass spectrometry analysis suggested that DDX5, a component of the microprocessor complex, acts as a VGLL3 interactor (Supplementary Fig. 12a), and we confirmed the interaction between endogenous VGLL3 and endogenous DDX5 (Supplementary Fig. 12b) in cardiac myofibroblasts. Importantly, similar to VGLL3 knockdown, DDX5 knockdown significantly increased miR-29b expression (Fig. 5p) and decreased collagen expression (Supplementary Fig. 12c), suggesting that VGLL3 regulates miR-29b production via DDX5.

9. Provide representative images and detailed description of Method section for echocardiogram, e.g. dosage of anesthesia and view axis, because the values in the Fig. 6e looks far from the standard average in anesthetized mice described in a previous review (Lindsey M.L. et al. *Am J Physiol Heart Circ Physiol.* 2018). Additionally, diastolic functions and/or stiffness should be evaluated to demonstrate that *Vgll3-KO* results in functional alterations due to less collagen deposition in the scar.

Response: As per your suggestion, we presented the representative images of the echocardiogram in Supplementary Fig.14f and described the detailed experimental conditions (including anaesthetic dosage and the view axis for echocardiogram measurement) in the Methods section of the revised manuscript (page 59, lines 15–16; page 60, lines 5, 6–10, and 11–13).

We learned the technique of echocardiography measurement from Prof. Hitoshi Kurose's laboratory. Thus, our echocardiogram values were similar to those reported in papers published by Kurose's lab and his related laboratory (Nakaya et al., *J Clin Invest.*, 2017; Horii Y et al., *FASEB J* 2020; Shimoda K et al., *Sci Rep*, 2020; etc.). We also found similar values in studies by others, such as the Prof Eric Olson group (for example, *J Clin Invest.* 2012;122(4):1222-1232).

However, in view of your concern, we changed the position of the probe against mouse hearts during echocardiogram measurement and remeasured the LVIDd, LVIDs, EF, and FS so that the values were similar to those of the standard average in the review (Lindsey M.L. et al. *Am J Physiol Heart Circ Physiol.* 2018) you presented. The results demonstrated that the values for EF and FS in the hearts 28 days after MI were significantly increased in *VGLL3 KO* mice, which is similar to the results of a previous study. The revised data was presented in Fig. 6g and Supplementary Table 1.

To follow up on your recommendation to evaluate diastolic function, we compared the diastolic function of WT and KO mice 28 days after MI. The ratio between the E- and A-waves (E/A ratio) is often used to assess diastolic function. Because our own echo equipment was not capable of measuring diastolic capacity, we borrowed a more sophisticated echo system capable of Doppler echocardiogram measurement. The latter demonstrated that the E/A ratio was greatly increased in mouse hearts 28 days after MI due to severe diastolic dysfunction. We found that the increase in the E/A ratio was significantly decreased in *VGLL3 KO* mice compared to that in WT mice (Supplementary Fig. 14g), suggesting that diastolic dysfunction after MI was more moderate in *VGLL3 KO* mice. This has been noted in our manuscript on page 29, lines 13–14, with the methodology described on page 60, lines 6–13.

Measuring the stiffness of cardiac tissue, as you have suggested, is a difficult task. To our knowledge, only a few groups have evaluated and compared differences in heart stiffness between

WT and KO mice. Most of these groups utilized an atomic force microscope (AFM). However, an AFM is an extremely expensive device not currently available to our research group; furthermore, we expect that it will take time to establish a hardness measurement system.

However, we speculated that the shear wave elastography method (normally used to measure fibrotic liver stiffness) could potentially be applied for the measurement of fibrotic heart stiffness. To test this, we employed a new ultrasound system with an SWE function of shear wave elastography. The measurement by the system demonstrated the stiffness of infarcted area in hearts after MI was about 18.2 kPa in WT mice and about 12.6 kPa in VGLL3 KO mice, as shown below in Fig. R5. This difference was statistically significant, and the values for stiffness were comparable to those measured by AFM in hypertrophic mouse hearts with fibrosis in previous reports (Ref #30: Hara et al., Circ Res. 2019 Aug 2;125(4):414-430).

Fig. R5 *Vgll3* deficiency in mouse attenuates the heart stiffness values on day 28 after MI surgery.

Stiffness of MI-operated hearts (WT, n = 9; KO, n = 8). Data are presented as mean \pm SEM. The *P* value was determined using the Mann-Whitney's *U* test, **P* < 0.05.

Thus, we believe that the stiffness of the fibrotic regions in the heart is reduced in *Vgll3* KO mice. However, we decided to include these data only in our response letter and not in the manuscript, firstly because this method is not well recognised in the field of cardiac fibrosis, unlike in the field of liver fibrosis. Secondly, we were aware of the reviewer's comment noting that diastolic function and/or stiffness should be evaluated, and we trust that our measurements on diastolic function will be sufficient.

10. It is unclear why the authors moved to in situ hybridization in vivo rather than immunohistochemistry when they clearly have an antibody that works. The images shown in figure 2 do not provide a good perspective on the extent of VGLL3 expression. Possibly, some quantification of the overall extent of overlap would be beneficial.

Response: We have found that the VGLL3 antibody used in our immunocytochemical

experiments (Fig. 1) was not applicable for immunohistochemistry. In addition, we found that no commercially available VGLL3 antibodies were applicable to immunohistochemistry. That was our reason for determining Vgll3 expression in fibrotic tissues using *in situ* hybridisation.

Regarding your concerns about Figure 2, we acquired images of various regions of the infarct area. Then, we have counted the expression of marker proteins of myofibroblasts (Fig. 2b), cardiomyocytes (Fig. 2c), and leukocytes (Fig. 2d) in more than 100 cells expressing *Vgll3* mRNA. The quantification data have been presented in each Figure.

11. The authors should discuss if there are any phenotypes in the VGLL3 knockout.

Response: We found no phenotypes in the VGLL3 KO mice under normal conditions. VGLL3 KO mice were born at the expected Mendelian ratios and did not display any overt phenotype during adulthood. However, in accordance with this comment, we mentioned the lack of phenotypes in *Vgll3* knockout mice in our Discussion section (page 31, lines 15–17): “It would be of interest to analyse female *Vgll3* KO mice in detail from this perspective, though we have found no phenotypes in the *Vgll3* KO mice under normal conditions.”

12. The authors should not state that cardiac function was improved unless they have the temporal data to support this statement. Line 11 page 26

Response: We agree with your assessment and, in accordance, our description was rewritten on page 29, lines 10–14.

Previous manuscript: “Consistent with these observations, echocardiography demonstrated that cardiac functions (ejection fraction and fractional shortening) were significantly improved in *Vgll3* KO mice 28 days after MI.”

Revised manuscript: “**Furthermore**, echocardiography demonstrated that **the values** for ejection fraction and fractional shortening rate, reflecting contractility, were significantly **increased** (Fig. 6g, Supplementary Fig. 14f, Supplementary Table 1) and E/A ratio, reflecting reduced diastolic function, was significantly decreased in *Vgll3* KO mice at 28 days after MI (Supplementary Fig. 14g).”

13. The authors need to justify that all statistical analyses were performed with parametric methods in this study. Otherwise non-parametric methods would be appropriate for the dataset without equality and normality.

Response: In accordance with this comment, we performed the Shapiro–Wilk test to determine

the distribution of variables for all datasets. Then, based on the Shapiro–Wilk test results, we could determine whether parametric or nonparametric methods were required. As a result, we changed the parametric methods to nonparametric methods for the analysis of some data. This implementation of the Shapiro–Wilk test is described in the Materials and Methods on page 62, lines 5–11: “We performed the Shapiro–Wilk test that determines the distribution of the variables for all data. Then, based on the respective Shapiro–Wilk test results, all data were analysed using parametric or nonparametric methods.”

Although the incorporation of the Shapiro–Wilk test sometimes changed the *P*-value of each data set, the results were basically the same as before.

14. The discussion has a great deal of results reiteration. Possibly, the authors can provide more of a summary rather than summarizing results a second time

Response: To address this concern, we have removed text that comprised a redescription of the results; instead, we included new discussion about the phenotype of *Vgll3* KO mice and the effect of substrate stiffness on the conditions of biomolecular condensates in the revised manuscript (page 31, line 13–page 32, line 13).

Minor comments

1. It is suggested that the authors provide more current reviews in their citations.

Response: In accordance with this comment, we have changed several old reviews to new ones, listed as reference:

#1 Tallquist, M. D. Cardiac Fibroblast Diversity. *Annu. Rev. Physiol.* 82, 63–78 (2020)

#2 Distler, J. H. W. et al. Shared and distinct mechanisms of fibrosis. *Nat. Rev. Rheumatol.* 15, 705–730 (2019)

#3 Henderson, N. C., Rieder, F. & Wynn, T. A. Fibrosis: from mechanisms to medicines. *Nature* 587, 555–566 (2020)

#4 Bochaton-Piallat, M.-L., Gabbiani, G. & Hinz, B. The myofibroblast in wound healing and fibrosis: answered and unanswered questions. *F1000Research* 5, (2016)

#5 Pakshir, P. et al. The myofibroblast at a glance. *J. Cell Sci.* 133, (2020)

#8 Frangogiannis, N. G. Transforming growth factor- β in myocardial disease. *Nat. Rev. Cardiol.* 19, 435–455 (2022)

2. The sentence in line 15, page 9 needs clarification. “Positive correlation” between differentiation by mechanical stress and Vgll3 expression level?

Response: Thank you for highlighting text requiring clarification. *VGLL3* mRNA expression is positively correlated with mRNA expression of collagens but not with differentiation in myofibroblasts. Thus, we have revised the sentence as follows (page 9, line 16– page 10, line 1): Previous manuscript: “*VGLL3* mRNA expression in myofibroblasts is positively correlated with their **differentiation** regulated by mechanical stimuli.”

Revised manuscript: “Expression of *Vgll3* mRNA in myofibroblasts or *VGLL3* mRNA in CCD-18Co is positively correlated with **mRNA expression of collagen** regulated by substrate stiffness (Supplementary Fig. 1j)”

We quantified the correlation between *VGLL3* mRNA expression and the mRNA expression of *Colla1* and *Colla2* and presented the data in Supplementary Fig. 1j.

3. Remove an equal sign in line 14, page 13 and correct the sentence.

Response: Thank you for pointing out this oversight. We have corrected this sentence as follows (page 14, lines 2–3):

Previous manuscript: “In contrast, *Vgll3* mRNA signals were absent in sham-operated control hearts (Supplementary Fig. 3e), = **largely devoid of** myofibroblasts.”

Revised manuscript: “In contrast, *Vgll3* mRNA signals were absent in the sham-operated control hearts (Supplementary Fig. 3e) **with few** myofibroblasts.”

4. Avoid the use of "correlate" unless two continuous variables are analyzed statistically (line 10, page 8 etc.). Otherwise quantify "mechanical stimulus-dependent myofibroblast differentiation" and the expression level of *Vgll3* followed by correlation analysis.

Response: We agree with your assessment. Because “myofibroblast differentiation” is not a continuous variable, we changed this description in relevant parts to ‘**expression of collagen production**’ (page 8, line 16) or ‘***Colla1* and *Colla2* expression**’ (page 15, line 4). We then examined the correlation between *VGLL3* and collagen expression levels. The results showed that expression levels were (strongly) correlated and have been included in Supplementary Fig. 1j and Supplementary Fig. 4b. Furthermore, we have removed the word "correlation" and modified the text (page 22, lines 2–3):

Previous manuscript: “These results demonstrated that the increased expression of collagens by *VGLL3* **is correlated with** its ability to undergo LLPS.”

Revised manuscript: “These results demonstrate that the increased expression of collagens by *VGLL3* **depends on** its ability to undergo LLPS.”

5. P 9 line 7 expression of this gene

Response: We have corrected this sentence as follows (page 9, lines 10–11):

Previous manuscript: “*Vgll4* was expressed in cardiac myofibroblasts (Supplementary Fig. 1d) but the expression of **these genes** was much lower compared to that of *Vgll3* (Supplementary Fig. 1e).”

Revised manuscript: “*Vgll4* was expressed in cardiac myofibroblasts (Supplementary Fig. 1f) but the expression of **this gene** was much lower compared to that of *Vgll3* (Supplementary Fig. 1g).”

6. No figure is referred to in the discussion of Rho/Rock signaling.

Response: Thank you for pointing out this oversight. We included references to the figures related to Rho/Rock signalling at the appropriate positions (page 12, lines 4–6):

“Treatment of cardiac myofibroblasts with the ROCK inhibitor Y27632 (**Fig. 1l, m**) or the Rho inhibitor cell-permeable C3 transferase (**Supplementary Fig. 2j, k**) significantly reduced VGLL3 nuclear localisation...”

7. Can the authors provide better examples of nuclear stains in figure 1K?

Response: Following your suggestion, we have replaced the images with new ones that display better staining of the nucleus (Fig. 1l in the revised manuscript).

REVIEWERS' COMMENTS

Reviewer #2 (Remarks to the Author):

The authors have substantially revised their manuscript, addressing many of the issues identified in a constructive manner.

In this reviewer's mind there still remains a question as to whether the very clear GFP-VGLL3 puncta are reflective of the much less obvious granular distribution of endogenous VGLL3. There is still the possibility that overexpression of GFP-VGLL3 results in artefactual formation of large phase-separated droplets which incorporate other phase separating proteins.

The micrographs including untagged endogenous protein, provide some evidence of colocalisation, however the statement on p19 that "endogenous VGLL3 formed discrete puncta in the nucleus, and these were co-localised with endogenous NONO" is a little stronger than Figure 4D supports (discrete puncta would be a way to describe 4F rather than 4D).

Nevertheless, colocalisation is demonstrated, and no longer attributed to specific organelles, and so the interpretation seems reasonable.

Reviewer #3 (Remarks to the Author):

The authors have addressed a majority of the major concerns.

One minor note is that the authors do not demonstrate improvement of ejection fraction. They demonstrate that EF is less impaired compared to controls.

Response to the reviewers' comments

Reviewer #2

The authors have substantially revised their manuscript, addressing many of the issues identified in a constructive manner.

In this reviewer's mind there still remains a question as to whether the very clear GFP-VGLL3 puncta are reflective of the much less obvious granular distribution of endogenous VGLL3. There is still the possibility that overexpression of GFP-VGLL3 results in artefactual formation of large phase-separated droplets which incorporate other phase separating proteins.

The micrographs including untagged endogenous protein, provide some evidence of colocalisation, however the statement on p19 that "endogenous VGLL3 formed discrete puncta in the nucleus, and these were co-localised with endogenous NONO" is a little stronger than Figure 4D supports (discrete puncta would be a way to describe 4F rather than 4D).

Nevertheless, colocalisation is demonstrated, and no longer attributed to specific organelles, and so the interpretation seems reasonable.

Response:

Thank you very much for your kind efforts on improving our manuscript.

As per your comments, we have revised the sentence (Page 19, line 16–17 in our previous manuscript) as follows:

Previous manuscript: “endogenous VGLL3 formed **discrete** puncta in the nucleus, and these were co-localised with endogenous NONO.”

Revised manuscript: “endogenous VGLL3 formed puncta in the nucleus, and these were co-localised with endogenous NONO.” (Page 19, line 16–17)

Reviewer #3

The authors have addressed a majority of the major concerns.

One minor note is that the authors do not demonstrate improvement of ejection fraction. They demonstrate that EF is less impaired compared to controls.

Response:

We are pleased to read the positive comment on our revised version and thank the reviewer for the careful comment.

In response to the comment, we have revised the legend of Figure 6 (Page 84, line 1–2 in our previous manuscript) as follows:

Previous manuscript: “*Vgll3* deficiency in mice attenuates cardiac fibrosis and **improves cardiac functions** after myocardial infarction.”

Revised manuscript: “*Vgll3* deficiency in mice attenuates cardiac fibrosis and **impairs cardiac dysfunctions** after myocardial infarction.” (Page 84, line 4–5)